# Modified Curcumins as Potential Drug Candidates for Breast Cancer: An Overview

**DOI:** 10.3390/molecules27248891

**Published:** 2022-12-14

**Authors:** Abigail L. Flint, David W. Hansen, LaVauria D. Brown, Laura E. Stewart, Eduardo Ortiz, Siva S. Panda

**Affiliations:** Department of Chemistry & Physics, Augusta University, Augusta, GA 30912, USA

**Keywords:** curcumin, curcumin mimic, conjugates, breast cancer, synthesis, drug development

## Abstract

Breast cancer (BC), the most common malignancy in women, results from significant alterations in genetic and epigenetic mechanisms that alter multiple signaling pathways in growth and malignant progression, leading to limited long-term survival. Current studies with numerous drug therapies have shown that BC is a complex disease with tumor heterogeneity, rapidity, and dynamics of the tumor microenvironment that result in resistance to existing therapy. Targeting a single cell-signaling pathway is unlikely to treat or prevent BC. Curcumin (a natural yellow pigment), the principal ingredient in the spice turmeric, is well-documented for its diverse pharmacological properties including anti-cancer activity. However, its clinical application has been limited because of its low solubility, stability, and bioavailability. To overcome the limitation of curcumin, several modified curcumin conjugates and curcumin mimics were developed and studied for their anti-cancer properties. In this review, we have focused on the application of curcumin mimics and their conjugates for breast cancer.

## 1. Introduction

Breast cancer is the most prevalent form of the disease found in women and is the main cause of cancer-related deaths in women globally. As with many other human cancers, breast cancer is caused by major modifications in genetic and epigenetic processes, as well as the targeting of many signaling pathways in the process of development and malignant progression toward an incurable and fatal disease. It has been established that a higher risk of breast cancer is related to both an earlier age at the onset of menarche as well as a later age at the onset of menopause. In the United States, roughly one in eight women at some point in their lives will be diagnosed with invasive breast cancer. This accounts for around thirteen percent of all women who are diagnosed with breast cancer. Furthermore, the Centers for Disease Control and Prevention (CDC) report breast cancer claims the lives of over 42,000 American women and 500 men each year [1,2]. As a result of this, research has been carried out to discover and develop drugs that are capable of effectively treating this particularly aggressive form of cancer. There is current difficulty in creating an effective treatment option for cancer, due to most available medications lacking the needed potency to provide full protection against the disease. The process of producing a new drug is time-consuming, challenging, and expensive. Additionally, it is intricate, and there is a high degree of unpredictability regarding the effectiveness of the drug after development. The difficulty in targeting cancer stem cells (CSCs), related to the drug-resistant properties of cancer stem cells rendering them immune to anti-cancer drugs, the lack of cancer epigenetic profiling, and the lack of specificity of existing epi-drugs are some of the most encountered challenges associated with cancer treatment [3].

Natural products have been used as a significant source of medications for many years. As of today, around half of all pharmaceuticals are still produced with the assistance of natural components. Several key commercialized medications in the field of cancer treatment have been derived from natural sources by structurally altering existing compounds or by synthesizing novel compounds like natural components. Research on modern anti-cancer drugs continues to place a significant emphasis on the search for improved cytotoxic agents. Due to the enormous structural diversity of natural compounds and the bioactivity potential of these compounds, several products that have been isolated from plants, marine flora, and microorganisms can serve as “lead” compounds. The improvement of their therapeutic potential through molecular modification can be carried out due to the large structural diversity that natural compounds possess [4].

Curcumin (Figure 1) is an example of a naturally occurring substance that has shown promise in treating breast cancer. Curcumin, a polyphenolic compound that can be found in turmeric (*Curcumin longa*), has been the topic of breast cancer research for over the past two decades due to its potential anti-inflammatory and anti-cancer effects. It has been revealed that curcumin suppresses the growth, spread, and metastasis of several malignancies. Its capacity to suppress oncogenic molecules such as protein kinases, cytokines, transcription factors, and growth factors plays a significant role in mediating its anti-cancer actions. In addition to this, curcumin obstructs the expansion and dissemination of cancer cells by obstructing their passage through the various phases of the cell cycle and/or by inducing apoptosis in cancer cells. However, according to pharmacokinetic studies, curcumin has poor systemic bioavailability because it is rapidly metabolized in the liver, where it undergoes glucuronidation and sulfation before being eliminated in the stool. The attempts undertaken up until this point to slow curcumin’s rapid metabolism have been ineffective in most cases. Due to curcumin’s restricted bioavailability and quick metabolism, researchers have previously explored and are currently exploring novel synthetic curcumin analogs that have lower toxicity, yet increased effectiveness [5,6].

Curcumin scaffolds have been extensively explored and investigated including several analogs, conjugates, and mimics to reform their pharmacological potency and improve bioavailability [7,8,9,10,11]. The scaffold of curcumin has been broken down to identify the potential sites for structural modifications. The structure–activity relationship of the curcumin-based compounds indicates the key structural components/modifications responsible for a specific target (Figure 2). Curcumin itself has many pharmacological properties such as modulating signaling molecules, including cytokines, chemokines, transcription factors, adhesion molecules, microRNAs, tumor suppressor genes, etc. Structural modifications of curcumin have been advocated for improving its bioavailability, enhancing stability, and increasing potency. The modified curcumin could serve as the next generation of drug candidates for cancer therapy. The present account elaborates on the literature reported on curcumin analogs, conjugates, and mimics as well their impact on anti-breast cancer properties.

Many review articles have been published on curcumin and its analogs for various biological applications [12,13,14,15,16,17], but the current review article aims to be a critical resource for the investigators involved or interested in curcumin and/or curcumin scaffolds for breast cancer. We have also determined the drug-like properties using a computational tool to analyze the drugability of the conjugates. It is also hoped that this review article will inspire researchers in natural product-based drug design and development.

## 2. Materials and Methods

A comprehensive and systematic review was conducted based on the databases of PubMed, Web of Science, Springer, and Scopus. The keywords “curcumin” AND “anticancer” were used and further filtered by “breast cancer”. The systematic search of electronic databases identified 1086 articles after excluding patents, clinical trials, and conference abstracts. We carefully reviewed all the articles and chose the articles which discuss and/or report on modified curcumins. The systematic portion of this review included 63 references after excluding non-relevant ones.

### 2.1. Anti-Breast Cancer Properties of Curcumin Analogs

The isoxazole curcumin analog **1** was synthesized and the anti-cancer properties against the MCF-7 breast cancer cell line and its multidrug-resistant (MDR) version, MCF-7R, were compared with curcumin. After 72 h of treatment, the IC_50_ of curcumin was calculated from four separate experiments to be 29.3 ± 1.7 μM in MCF-7 and 26.2 ± 1.6 μM in MCF-7R, indicating that the cytotoxic activity of curcumin in the MDR breast cancer cell line is at least equivalent to, and perhaps slightly stronger than, its parental variant. In both the parental and MDR cell line, derivative **1** was more effective than curcumin with an IC_50_ of 13.1 ± 1.6 μM in MCF-7 and 12.0 ± 2.0 μM in MCF-7R. An MDR form of HL-60 leukemia also showed comparable outcomes. RT-PCR analyses in MCF-7 and MCF-7R cell lines revealed that curcumin and **1** caused early changes in the quantities of important gene transcripts, which were, nevertheless, primarily varied between the two cell lines. Overall, these results show that the expression of P-gp or the absence of ER in breast cancer cells does not impede the anti-cancer activities of either curcumin or **1**. Remarkably, the agents seemed to adjust their molecular actions in response to the different patterns of gene expression found in the MDR and the parental MCF-7 [18].



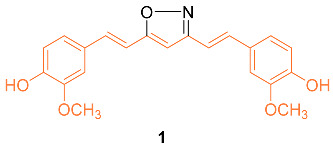



According to Wang et al., research was conducted on hydrazinocurcumin **2** (HC) to investigate its effectiveness against breast cancer cells, specifically in the cell lines MDA-MB-231 and MCF-7. After 72 h of treatment, dose-dependent suppression of tumor cell survival and proliferation was seen for the MDA-MB-231 and MCF-7 cell lines. The IC_50_ values for **2** were 3.37 μM and 2.57 μM, respectively, which were both significantly lower than those for curcumin (26.9 μM and 21.22 μM). Compared to curcumin, the results demonstrated that **2** was significantly more effective in suppressing cell viability in both cell lines tested. Apoptosis was induced in MDA-MB-231 and MCF-7 cells using FCM, and the influence of **2** and curcumin on this process was analyzed. At 10 µM, **2** significantly induced cells apoptosis (14% in MDA-MB-231 cells and 26% in MCF-7 cells), whereas at the same concentration, curcumin only induced 9% and 20% cell apoptosis in MDA-MB-231 and MCF-7 cells, respectively. The results showed that **2** caused an increase in the apoptotic rate of cells in a dose-dependent manner after a treatment period of 48 h. In addition, the Western blot analysis demonstrated that **2** was much more effective than curcumin in suppressing the production of STAT3 protein in MDA-MB-231 and MCF-7 cells at the same concentration (10–20 μM). The data showed that **2** was more effective than curcumin at suppressing cell proliferation, losing colony formation, depressing cell migration and invasion, and inducing cell death via inhibiting STAT3 phosphorylation and downregulating an array of STAT3 downstream targets [19].



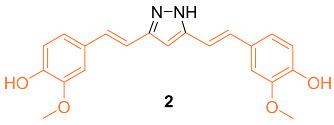



Mohankumar et al. studied the apoptotic mechanism of **3**, an *ortho*-hydroxy substituted analog of curcumin using an in vitro and in silico approach. In the study, it was found that **3** exhibited a greater potency in the modulation of selective apoptotic markers and inhibited MCF-7 at a dose level of 30 µM (equivalent dosage level to curcumin), and significantly regulated PI3k/Akt, both intrinsic and extrinsic apoptotic pathways, by inhibiting Bcl-2 and inducing p53, Bax, cytochrome c, Apaf-1, FasL, caspases-8, 9, 3, and PARP cleavage. mRNA expression studies for Bcl-2/Bax indicated increased efficiency with **3** compared to curcumin, while an in silico molecular docking study utilizing PI3K revealed that the docking of **8** was more potent than curcumin. Cells treated with **3** effectively induced apoptosis through ROS intermediates, as measured by 2′,7′-dichlorodihydrofluorescein diacetate (DCFH-DA). Results showed **3** induced apoptosis more effectively than curcumin, and this activity can be attributed to the presence of the hydroxyl group in the *ortho* position in the structure [20]. In addition, Western blotting indicated that compound **3** significantly downregulated the expression levels of NF-κB, p65, and c-Rel. In addition, src levels were significantly reduced in comparison to cells treated with curcumin. In silico docking studies were performed with the derivative and curcumin with NF-κB (PDB ID: 1NFK). The results indicated that the derivative displayed a stronger interaction with NF-κB compared to curcumin, with a Lidblock score of 109.814 while curcumin’s was 95.696 [21]. 



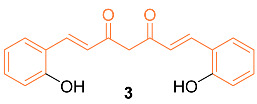



Lien et al. synthesized over 30 curcumin derivatives and published findings that a novel curcumin derivative (**4**) inhibits cell proliferation and drug resistance of HER2-overexpressing cancer cells. The mimic was tested in vitro on both the MCF-7 and MDA-MB-435 cell lines transfected with pSV2-*erbB2*. Results indicated that the derivative preferentially suppresses the growth of HER2-overexpressing cancer cells. Studies were also carried out to investigate if the derivative would sensitize HER2-overexpressing cancer cells to clinical drugs and it was found that overexpressing cells showed greater cytotoxic activity when the derivative was administered in combination with doxorubicin (DOX), etoposide, or taxol [22].



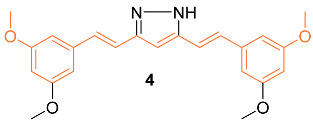



To understand the molecular hybridization impact and the integration of two drugs with different modes of action, affecting the same target, a variety of heterocyclic steroids and curcumin moieties were considered for the synthesis of hybrid conjugates and to determine their anti-cancer activity. The authors synthesized the hetero-steroid compounds and conducted in vitro studies of the cytotoxic effects against the MCF-7 breast cancer cell line. Of all compounds, **5** had the best cytotoxic activity against the MCF-7 cell line with an IC_50_ value of 18 μM. This compound is also promising as an anti-cancer compound having pro-apoptotic effects resulting in desired cell growth inhibition [23].



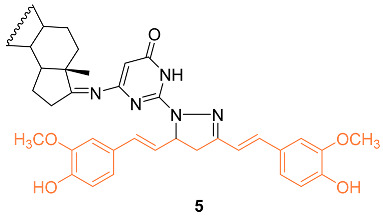



Bhuvaneswari et al. reported the biological evaluation and molecular docking of novel curcumin derivatives **6a**–**l** and **7a**–**k**. Firstly, in vitro cytotoxicity was tested against the MCF-7 breast cancer cell line. The IC_50_ for **6j** and **7i** were 15 µM and 10 µM, respectively. When the compounds were tested against normal HBL-100 cells, the cells were resistant to the compounds up to 50 µM doses, showing the compounds are selective and dose-dependent. Molecular docking studies were also conducted in PatchDock and suggested that these two compounds could be the starting point for designing new potent Bcl-2 anti-apoptotic protein inhibitors, with **7** having a geometrical score of 6028 and **6** with a score of 5962 compared to 4190 for curcumin (PDB: 1GJH) [24].



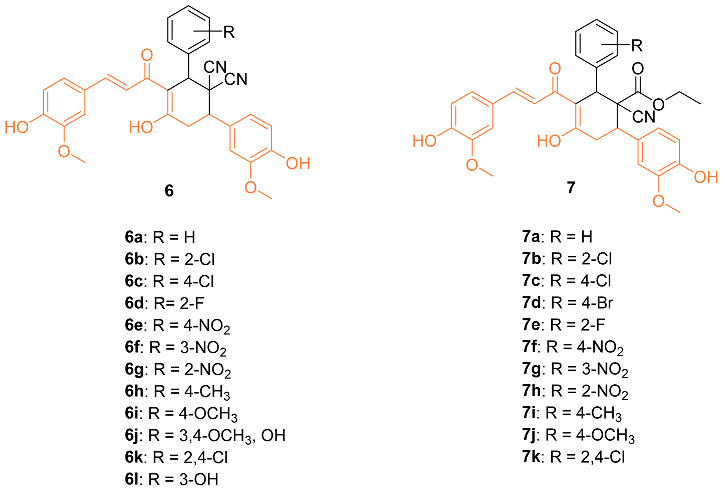



Nagwa et al. synthesized a set of curcumin derivatives **8a**–**g** and then experimented to determine the efficacy of the derivatives against breast cancer. Preliminary tests were conducted with normal MCF-10A cells and it was found that all derivatives had little cytotoxicity, with more than 85% cell viability. An MTT assay was performed with the derivatives against an MCF-7 breast cancer cell line. Compounds **8a** and **8c** were the most potent against the breast cancer cells, with an IC_50_ of 20 and 22 μg/mL, respectively. Pharmacokinetic (ADME) studies confirm that compounds **8a** and **8c** have good intestinal absorption and are non-carcinogenic [25].



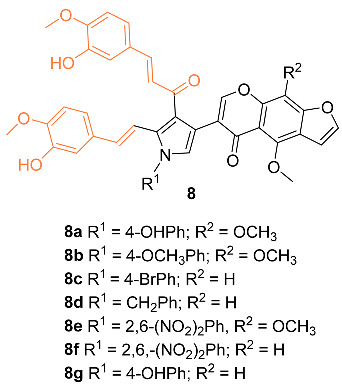



Hong et al. reported the synthesis of and anti-cancer studies on the novel curcumin mimic (1*E*,4*E*)-1,7-bis(4-hydroxyphenyl) (hepta-1,4-dien-3-one) **9** isolated from mistletoe. It was first tested for in vitro cytotoxicity in which it showed activity in the micromolar range. Additionally, **9** showed a higher potency than cis-platinum against four human breast cancer cell lines (SKBR3, MDA-MB231, MCF-7, and MDA-MB453). The IC_50_ values for the breast cancer cell lines were significantly lower with **9** compared to cis-platinum. The cytotoxicity of **9** with normal cells was investigated with LO2 human liver cells, GES-1 human gastric epithelial cells, and BEAS-2B human lung epithelial cells. The results indicated that **9** had a little inhibitory effect on normal cells, with each group having a less than 5% inhibition rate, which is much lower than the rate on cancer cells at the same concentration, indicating **9** has a selectivity for the toxic effects of cancer cells rather than normal cells. In addition, in vivo data on the MCF-7 breast cancer model in mice suggest that **9** is more effective than cisplatin. The groups administered **9** had a stable weight for up to 9 days, while a clear weight loss was observed in the positive control group [26].



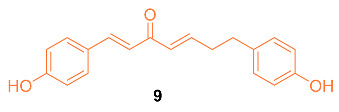



Shen et al. tested the efficacy of a curcumin analog **10** in breast cancer cells. The breast cancer cell lines MCF-7 and MDA-MB-231 were used to study the cell viability, cell migration, cell cycle, and apoptosis of this analog. It was shown that when the concentration of **10** was increased, there was a decrease in cell viability. In addition, **10** had an IC_50_ of 8.84 μM compared to curcumin with an IC_50_ of 16.85 μM against MCF-7 breast cancer cells. It was shown that **10** is a compound that activates the mitochondrial apoptosis pathway in breast cancer cells [27].



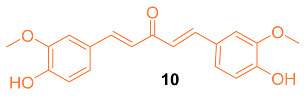



Sharma et al. synthesized 3,4-Dihydropyrimidin-2(1H)-one/thione curcumin analogs and, among them, compounds **11a**–**c** were submitted to the National Cancer Institute (NCI) to investigate activity against various cell lines, including the breast cancer cell lines MDA-MB-231 and HS 578T. At a concentration of 100 µM, compounds **11a**–**c** all displayed moderate activity, with compound **11a** being the most active. This is supported by a growth percent value (GP) of 55.45 for compound **11a** on MDA-MB-231 cells and a GP of 73.39 on HS 578T cells, while activities with a GP of 73.63 and 67.70 on MDA-MB-231 cells were found for compounds **11b** and **11c**, respectively. The authors believe compounds **11a**–**c** should be further studied to increase the moderate anti-cancer activity [28].



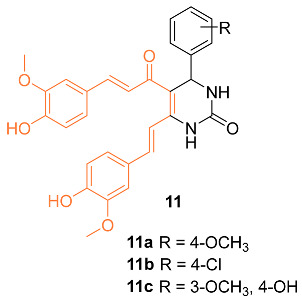



Zhang et al. reported on the synthesis and anti-cancer activity of ten curcumin mimics **12a**–**j**. Compound **12b** exhibited the best anti-cancer activity, with an IC_50_ value of 4.99 µM against MDA-MB-231 breast cancer cells compared to the 6.18 µM of cisplatin. In vivo data were obtained and were promising. However, in vivo testing was only carried out on H22 hepatic cells. Further testing is needed to evaluate if compound **12b** is a promising anti-breast cancer drug [29].



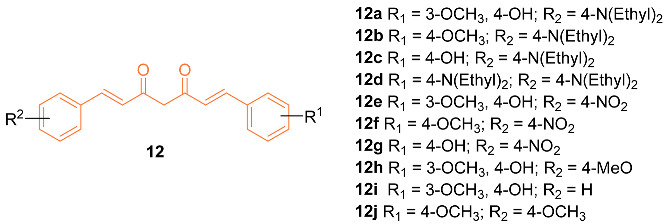



Considering the importance of pyrazole moiety, Ahsan et al. synthesized various curcumin analogs **13**–**15** containing a pyrazole or pyrimidine ring to target the epidermal growth factor receptor (EGFR) tyrosine kinase. Fourteen curcumin analogs with pyrazole or pyrimidine moieties were synthesized, with ten being evaluated amongst 60 different cell lines to observe anti-cancer effects. The activity was observed from various compounds, however, **13**–**15** displayed anti-cancer activity against various cell lines including MDA-MB-468. Compound **13** showed a cell promotion of −30.34%, compound **14** showed −31.86%, and compound **15** showed −35.04%. Ahsan et al. claim their curcumin analogs are promising and can be a therapeutic intervention in cancer treatment [30].



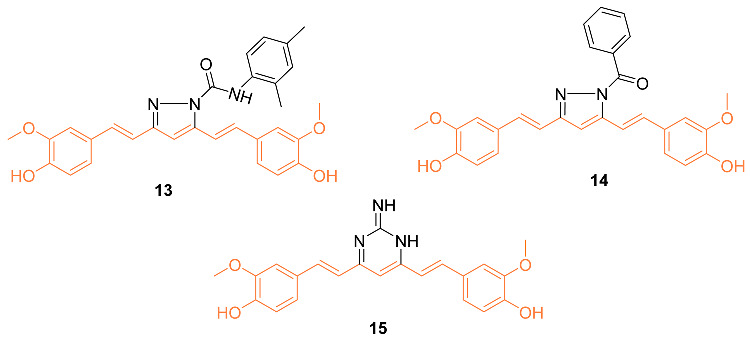



A set of twenty-four different analogs of curcumin containing pentadienone moiety were synthesized and examined for their anti-cancer properties against breast cancer cells (MCF-7 and MDA-MB-231). A dose-dependent suppression of tumor cell survival and proliferation was observed after 72 h of treatment with compounds **16**–**18**. The IC_50_ values for compound **16** were 2.7 ± 0.5 μM and 1.5 ± 0.1 μM for the cell lines MCF-7 and MDA-MB-231, respectively, which were 5-8 times lower than those for curcumin (21.5 ± 4.7 μM and 25.6 ± 4.8 μM). Furthermore, the IC_50_ values of compounds **17** (0.4 ± 0.1 μM and 0.6 ± 0.1 μM) and **18** (2.4 ± 1.0 μM and 2.4 ± 0.4 μM) were favorable for the MCF-7 and MDA-MB-231 cell lines, respectively. The non-malignant mammary epithelial cell line (MCF-10) demonstrated no toxicity from any of the three compounds. In comparison to curcumin, which did not exhibit any selectivity against cancer cell lines, it was discovered that compounds **16**–**18** displayed a selectivity ratio of at least fivefold or greater. Compound **17**, having IC_50_ values in the sub-micromolar range and a selectivity ratio greater than 25, was found to be the most potent analog. All three compounds, however, show promise as possible anti-tumor drug candidates for breast cancer [31].



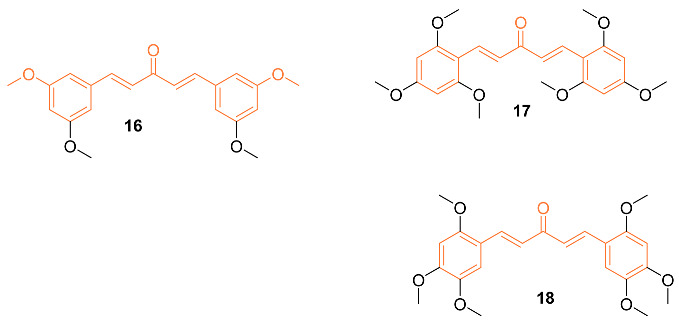



Ali et al. synthesized curcumin analogs **19a**, **19b**, and **20** and tested them against several breast cancer lines to determine their anti-cancer effects. The compounds were docked against the epidermal growth factor receptor, which allowed for the determination of binding efficiency. All derivatives showed moderate inhibition of epidermal growth factor receptors. Compounds **19b** and **20** showed the most anti-cancer activity against BT-549 with GI_50_ values of 2.98 μM for 2 and 1.51 μM for 3 [32]. 



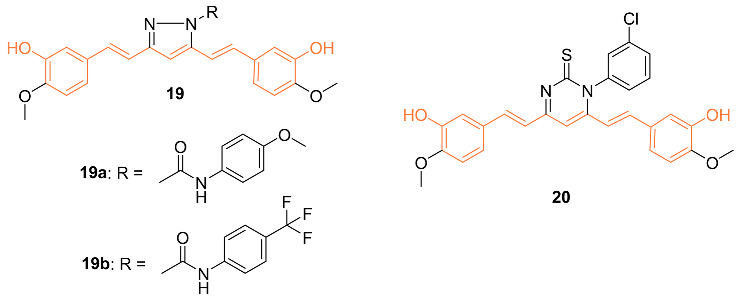



Besides the anti-cancer properties against breast cancer cell lines, some of the potential analogs were also further investigated to understand the mechanisms involved in the inhibition. We have summarized the possible pathways, molecular targets, and cell cycle phase arrest in Table 1.

### 2.2. Anti-Breast Cancer Properties of Curcumin Conjugates

Basnet et al. reported that diacetylcurcumin **21** (DAC), a synthetic derivative of CUR, showed strong nitric oxide (NO) and O_2^−^_ anion scavenger properties. However, this modification did not solve the basic problem of low aqueous solubility and poor bioavailability, which limits further studies [33]. Mehranfar et al. combined DAC with bovine casein nanoparticles (B-CNs) (Figure 3) to determine anti-cancer effects and improve bioavailability. DAC with Fe (III) is more stable due to increased basicity and additional anti-bacterial effects. In vitro anti-cancer studies performed on MCF-7 human breast cancer cells suggest the DAC-β-casein complex, creating a nanoformulation, displayed greater cytotoxic effects with an IC_50_ value of 22.5 µM compared to DAC (IC_50_ = 26.5 µM). The authors claim that B-CN micelles interact mainly via hydrophobic interactions that increase the solubility, bioavailability, and anti-tumor activity of DAC [34].

Another attempt has been to facilitate curcumin delivery to mitochondria by conjugating curcumin with lipophilic triphenylphosphonium (TPP) cation. The authors synthesized three mitocurcuminoids **22**–**24** by tagging TPP to curcumin at different positions, verifying uptake in mitochondria using ESI-MS analysis. With analysis showing significantly higher uptake of the mitocurcuminoids in mitochondria compared to curcumin in MCF-7 breast cancer cells, the mitocurcuminoids exhibited cytotoxicity to MCF-7, MDA-MB-231, SKNSH, DU-145, and HeLa cancer cells with minimal effect on normal mammary epithelial cells (MCF-10A). The study found that **22** and **24** accumulated most significantly in the mitochondria of MCF-7 cells compared to **23**; this was accredited to the presence of two TPP moieties in these molecules. Mitocurcuminoids often had a lower IC_50_ when compared to curcumin in various cancer cells using an SRB assay; curcumin ranged from 37.87 µM to 50 µM, while the mitocurcuminoids ranged from 2.31 µM to 8.62 µM. The mitocurcuminoids induced significant ROS generation, a drop in ∆Øm, cell cycle arrest, and apoptosis while inhibiting Akt and STAT3 phosphorylation, increasing ERK phosphorylation, and upregulating pro-apoptotic BNIP3 expression [35].



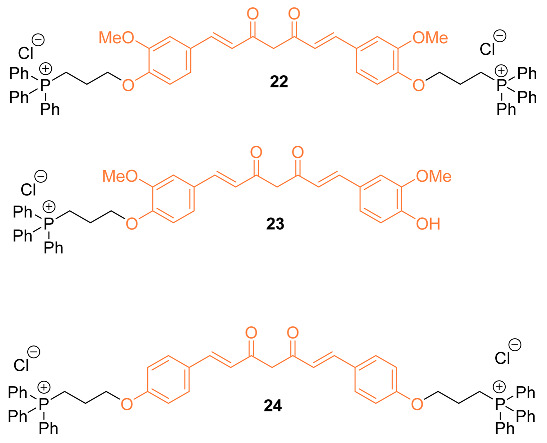



Recently, organometallic compounds gained attention for the development of potential drug candidates for cancer therapy. Among various organometallic compounds, copper complexes are more frequently used as cancer chemotherapeutics because of their selective inhibitory properties against topoisomerases. Additionally, copper complexes are essential in various biological processes such as mitochondrial respiratory reactions, cellular stress response, energy generation, and other body functions. Copper (II) complexes based on curcumin derivatives, **25**, **26** (**25** = 1,7-bis [4-(2-oxymethylenepyridine)-3-methoxyl]phenyl-1,6-heptadiene-3,5-diketone and **26** = 1,7-bis [4-(3-oxymethylene-2-chlorothiophene)-3-methoxyl] phenyl-1,6-heptadiene-3,5-diketone) were investigated for cytotoxic and DNA binding properties against various cell lines including MCF-7 breast cancer cell lines. As a result, **25** and **26** both showed greater cytotoxic activity than cisplatin. This was supported by IC_50_ values at 24 h of 44.51 ± 1.74 µM and 49.62 ± 2.23 µM, while at 48 h of 42.43 ± 1.63 µM and 46.32 ± 1.68 µM, respectively [36].



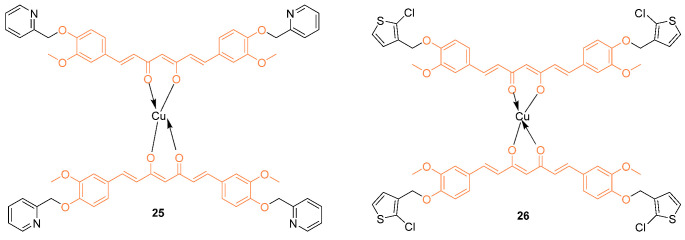



Hsieh et al. synthesized novel bis(hydroxymethyl) alkanoate curcuminoid derivatives **27a**,**b** and evaluated their activity against breast cancer in vitro and in vivo. For the in vitro studies, compound **27a** was significantly more potent against MDA-MB-231, with an IC_50_ of 2.67 µM compared to curcumin’s 16.23 µM, a 6-fold increase. When methyl groups were added to the compound to form **27b** and prevent tautomerization, the IC_50_ value decreased to 1.98 µM. Compound **27a** was then tested for inhibitory activity against ER^+^/PR^+^ breast cancer (MCF-7, T47D), and HER 2^+^ breast cancer (SKBR3, BT474, and MDA-MB-457). Compound **27a** exhibited significant inhibitory activity, with potencies 2.5–8.6 times greater than curcumin. Inhibitory effects for both **27a** and **27b** were further studied with a doxorubicin-resistant MDA-MB-231 cell line. Compound **27a** with an IC_50_ of 6.5 µM and compound **27b** with an IC_50_ of 5.7 µM showed ten-fold greater activity than curcumin, which had an IC_50_ of 57.6 µM. In addition, a synergistic effect was observed when these analogs were administered alongside doxorubicin. Lastly, in vivo data were gathered with compound **27a** in the MDA-MB-231 xenograft nude mouse model with oral doses of 5, 10, 25, and 50 mg/kg doses daily. Significant anti-tumor activity was observed at just 5 mg/kg a day, and at 10 mg/kg a day, it was equivalent to a 100 mg/kg daily dose of curcumin. Increasing the dose to 50 mg/kg daily led to a tumor size reduced by 60%. No adverse effects were observed in the mice upon study completion. An additional study was completed by administering doxorubicin at 50 mg/kg and 1 mg/kg daily, respectively, and tumor size was reduced by up to 80%. The cardiotoxicity also decreased compared to using doxorubicin alone [37].



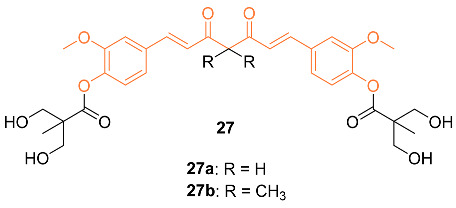



Lanthanide complexes gained importance in cancer diagnosis and treatment due to their versatile chemical and magnetic properties of the lanthanide-ion 4f electronic configuration. The novel lanthanide (III)–curcumin complex **28** was studied as a possible photocytotoxic agent against breast cancer. The compound showed an IC_50_ of 1.9 ± 1.2 μM when exposed to visible light in the incubation process against MCF-7 cells, while the complex was non-toxic when incubated in the dark. The complex was also non-toxic to MCF-10A normal cells in both conditions [38].



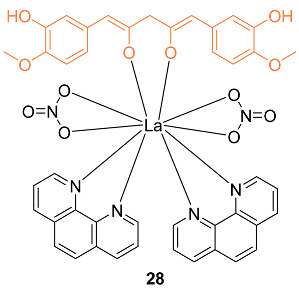



Bai et al. reported the synthesis of the pectin–curcumin conjugate **29** and studied its efficacy as a possible treatment for breast cancer. Cytotoxicity studies were performed on human MCF-7 cells in which the conjugate had an IC_50_ value of 12.0 ± 3.0 µM compared to 48.3 ± 2.9 µM of curcumin. In addition, the conjugate was shown to be less cytotoxic to normal cells than free curcumin. Critical micelle concentration for the conjugate was then found to be 0.04 mg/mL [39].



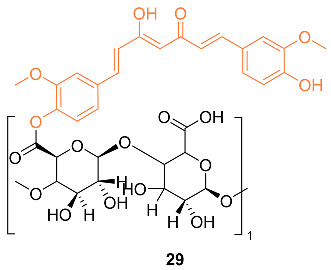



Bonaccosi et al. analyzed five curcumin derivatives with some involving the substitution of the phenolic OH groups. The cytotoxic effects of the five synthesized curcumin derivatives were tested against two triple-negative breast cancer cell lines, SUM 149 and MDA-MB-231. Compounds **30**–**32** showed greater cytotoxic effects against the SUM 149 cells experiencing the greatest sensitivity. Analysis of the SUM 149 cell line showed IC_50_ values of 11.2 ± 1.30 μM, 13.2 ± 1.59 μM, and 13.5 ± 0.88 μM, while the IC_50_ values against the MDA-MB-231 cell line are 18.0 ± 0.41 μM, 20.0 ± 0.00 μM, and 15.0 ± 0.85 μM for compounds **30**–**32**, respectively. These compounds proved to be cytotoxic, while further studies by the authors showed pro-apoptotic abilities and interference with the NF-κB pathway, making these derivatives possible anti-cancer agents [40].



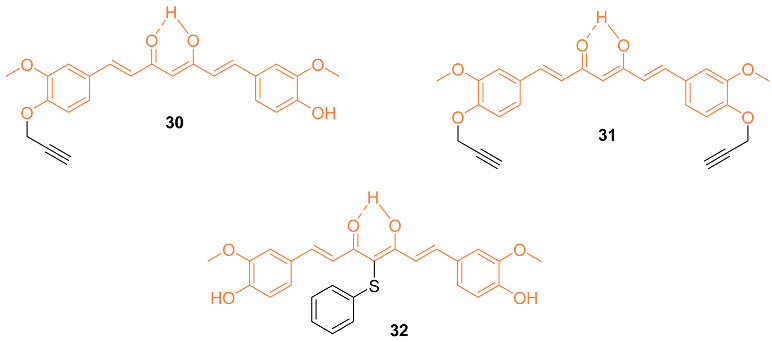



According to Sertel et al., over 60 distinct cell lines were used in their study to test curcumin derivatives **33**–**36** for their cytotoxicity and IC_50_ values. During the process of determining the IC_50_ values, these derivatives were subjected to testing over a dose range of 10^−8^ to 10^−4^ M using a variety of cell lines including breast cancer cell lines. When compared to the derivatives, it was found that curcumin had the lowest level of activity, while derivative **36** had the highest level, with the remaining derivatives showing intermediate cytotoxicity. In addition, the curcumin-like compounds were found to not contribute to ATP-binding cassette transporter-mediated multidrug resistance and to have no effect on the sensitivity of cancer cells to conventional anti-cancer medicines. Not only does this class of compound possess a high level of cytotoxic activity, but it also lacks cross-resistance to 1400 different standard agents and is not involved in ABC transporter-mediated multidrug resistance. This combination of properties makes these derivatives intriguing candidates for the treatment of cancer [41].



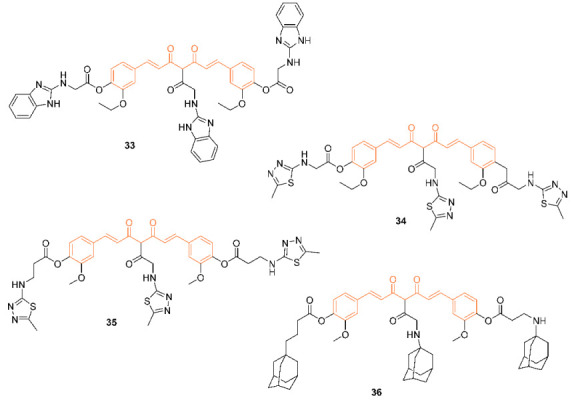



Kesharwani et al. reported on the synthesis and activity of five curcumin analogs **37a**–**e.** A docking simulation was performed with curcumin and the five analogs with Aldehyde dehydrogenase 1 (ALDH1A1) and glycogen synthase kinase-3 beta protein (GSK-3β), both of which are upregulated and overexpressed in breast cancer cells. Compounds **37a**,**b**,**e** had an improved binding affinity compared to curcumin, with G scores of −11.83, −11.31, and −11.79, respectively, compared to curcumin’s −10.36 in ALDH1A1. Compound **37e** also had a superior G score compared to curcumin when tested for binding affinity with GSK-3β, having a score of −10.91 compared to curcumin’s −10.62. Antioxidant results also indicated that compound **37e** had superior antioxidant activity to curcumin due to the four phenoxy groups [42].



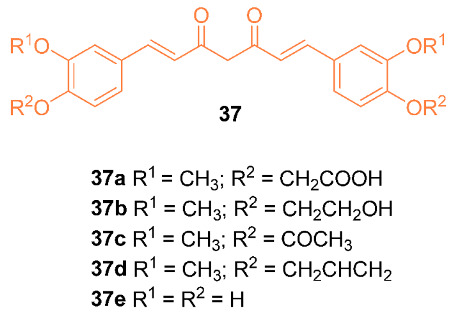



Upadhyay et al. reported on the synthesis and biological activity of platinum (II) curcumin complexes **38a**–**c**. In vitro testing was carried out against A549, HeLa, MDA-MB-231, and HPL1D cell lines. IC_50_ values were obtained with three separate methods: (1) in the dark with 4 h of preincubation, and 20 h of postincubation; (2) 4 h of preincubation in the dark followed by subsequent exposure to visible light of 400–700 nm and 19 h of postincubation in the dark; (3) 4 h of preincubation in the dark followed by exposure to red light of 600–720 nm and 19 h of postincubation in the dark. The first method showed IC_50_ values of 63.3 μM, 57.3 μM, and 49.6 μM for MDA-MB-231 and 72.6 μM, 66.4 μM, and 64.7 μM for HPL1D cells for compounds **38a**–**c**, respectively. In the second method, the IC_50_ values were 15.6 μM, 11.3 μM, and 9.5 μM for MDA-MB-231, and 18.2 μM, 15.6 μM, and 10.8 μM for HPL1D cells for compounds **38a**–**c**, respectively. The final method was only tested on compounds **38b**,**c** in which the IC_50_ values were 6.9 μM and 2.4 μM for MDA-MB-231 cells and 7.5 μM and 3.7 μM for HPL1D cells for compounds **38b** and **38c**, respectively. Flow cytometry studies indicated that the compounds caused cell cycle arrest at the sub-G_1_ phase via apoptosis [43].



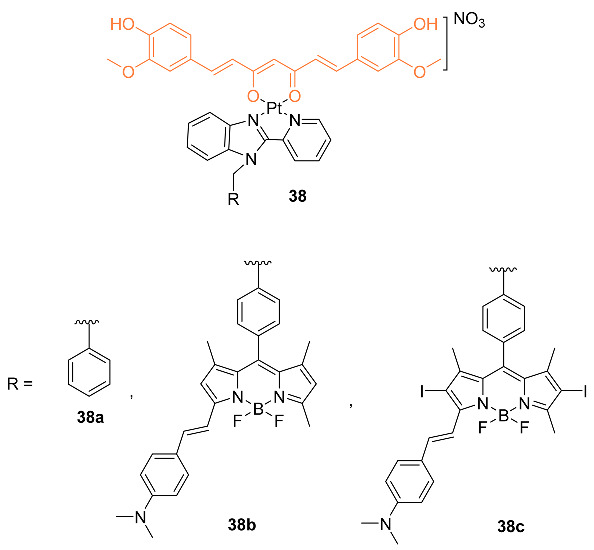



According to Panda et al., to circumvent their limited bioavailability and unfavorable side effects, conjugates of curcumin (CUR) and dichloroacetate (DCA) **39**, **40a**–**e** were made utilizing a molecular hybridization technique. Molecular docking was used in the research, and the results showed that breast cancer cells include potential targets for curcumin-modified conjugates. It was discovered that curcumin had a binding energy of around −24 kJ/mol, a ligand efficiency of approximately 0.22, and a docking score of approximately −29.24. On the other hand, **40a** displayed a substantially greater range of these characteristics, which indicates that this molecule is likely to inhibit DYRK2. These six conjugates did not show any significant levels of toxicity in either an in vitro test utilizing a human normal immortalized mammary epithelial cell line (MCF-10A) or an in vivo test using C57BL/6 mice. However, colony-forming assays demonstrated that treatment with **39** and **40a** dramatically suppressed both the growth and clonogenic survival of several human BCs. The treatment of a transgenic mouse with breast cancer and mice with metastatic breast cancer tumors with **40a** through oral gavage greatly reduced tumor growth and metastasis. Overall, the result of this research provides compelling evidence that CUR and DCA conjugates have significant anti-cancer properties even at concentrations below the micromolar threshold. These conjugates also overcome a clinical limitation associated with the use of CUR and DCA as potential anti-cancer drugs [44].



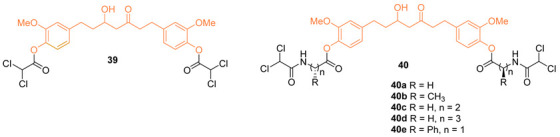



The mechanistic studies of some potential curcumin conjugates are reported. It is interesting to learn the impact of the conjugation of curcumin on the mode of action, upregulation/downregulation, and phases of cell cycle arrest. In Table 2, we have summarized the available information on the curcumin conjugates which we included in this section.

### 2.3. Anti-Breast Cancer Properties of Curcumin Mimic Conjugates

Lin et al. investigated the underlying mechanisms of dibenzoylmethane **41** (DBM), a curcumin-diketone analog, in terms of chemopreventive activities in mammary tumorigenesis. Studies on competitive estrogen receptor binding show that DBM directly binds to estrogen receptors in vitro, causing apoptosis in a variety of cancer cells during Phase I/Phase II metabolic systems. In vivo proliferation studies revealed that DBM may play an anti-estrogenic role. According to the findings of these studies, DBM significantly reduces the expression of *bcl-2*, *c-myc, Ha-ras*, and *hTERT* and inhibits E_2_-induced proliferation in both a mouse model and the human breast cancer cell line MCF-7. By considering the ER-ERE binding in these oncogenes’ regulatory regions, it was confirmed that DBM functions as an anti-estrogenic substance and can be a better alternative for curcumin [45]. The authors investigated the underlying mechanisms of DBM in the prevention of mammary tumorigenesis as an effector of the Phase I enzymatic system. This study investigated the effects of 1,12-dimethylbenz[a]anthracene (DMBA) on mammary tumorigenesis in Sencar mice. In an NADPH-dependent mix function oxidase system, the HPLC profile for DBM metabolism by liver microsomes from Sencar mice fed a control AIN-76A diet or 1% DBM diet was studied. According to the findings, the formation of a more polar reductive metabolite than DBM will facilitate conjugative metabolism by Phase II enzymes and subsequent excretion. Additionally, it was found that cytochrome P450, NADPH cytochrome P450 reductase, and phosphatidylcholine are the two protein and lipid components of the cytochrome P440-dependent drug metabolism system. The major reductive metabolite of DBM, DBMH2, was isolated, and several other minor metabolites were identified by NMR, GC, and LC-MS. This information clarified the potential function of DBM as a modulator of the cytochrome P450 reductase, which is necessary for the activity of the oxidase to metabolize DMBA. This leads to the conclusion that DBM has a remarkable inhibitory effect on 7,12-dimethylbenz[a]anthracene (DMBA)-induced mammary tumorigenesis, as well as on DMBA-induced mouse mammary tumorigenesis [46].



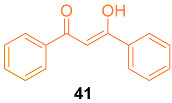



It is well known that prolonged endoplasmic reticulum (ER) stress can activate apoptosis. Recently, curcumin was identified to exert its pro-apoptotic effects by inducing ER stress in several tumor cells, including acute promyelocytic leukemia cells, human non-small cell lung cancer H460 cells, and human liposarcoma cells. Liu et al. used these advancements in curcumin to further investigate ER stress-mediated mechanisms and synthesized several mono-carbonyl analogs of curcumin (MACs). Of the seven active MACs, **42** displayed the strongest anti-tumor activity against various cancer cell lines including MCF-7 cells with an IC_50_ value of 2.21 ± 0.21 μM. The authors claim improved anti-cancer properties of the synthesized MACs in both in vitro and in vivo studies. However, the molecular mechanism is unclear [47].



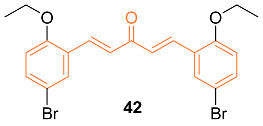



Meiyanto et al. explored the effects of curcumin and its two analogs, **43** and **44**, alone and in combination with doxorubicin to determine the effect on MCF-7/Dox cells featuring over-expression of HER2. Assays during the study indicated cytotoxic effects against MCF-7/Dox with IC_50_ values of 80 µM for curcumin, 21 µM for **43**, and 82 µM for **44**. When used in conjunction with doxorubicin, there is an increase in MCF-7/Dox sensitivity. This was verified using cell cycle distribution analysis; the combination caused an increase in sub-G1 cell populations. However, curcumin and **44** also decreased the localization of p65 into the nucleus induced by doxorubicin, indicating the inhibition of HER2 activity and NF-κB activation. Docking models indicated that all three compounds had a high binding affinity to HER2 and IKK, with curcumin and **44** having similar affinity levels towards proteins involved in the proliferative signal, including HER2, EGRF, IKK, and ER in both in vitro and in silico studies. It was discovered that **44** outperformed the other compounds in cytotoxic activity against MCF-7/Dox even at a low dose [48].



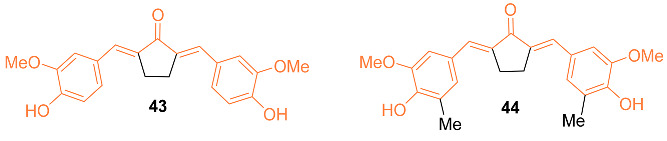



Monocarbonyl curcumin derivatives have previously shown an array of pharmacological activities in previous studies, however, many of these analogs are symmetric. With limited prior reports, Li et al. further investigated the pharmaceutical properties of asymmetric curcumin analogs. The authors synthesized twelve new asymmetric and five symmetric curcumin analogs **45a**–**p**. These compounds were tested against three cancer cell lines including MCF-7. The synthesized compounds showed greater cytotoxicity than curcumin, with compound **45o** showing selective cytotoxicity to the MCF-7 cells. Compounds displayed IC_50_ values of 26.62 μM, 39.94 μM, 13.90 μM, 22.53 μM, 21.70 μM, 67.10 μM, 40.3 μM, 56.54 μM, 28.94 μM, 41.9μM, 33.32μM, 22.15 μM, 40.71 μM, 52.77 μM, 9.11 μM, and 43.60 μM, for compounds **45a**–**p**, respectively, compared to 70.20 μM for curcumin against MCF-7 cells [49].



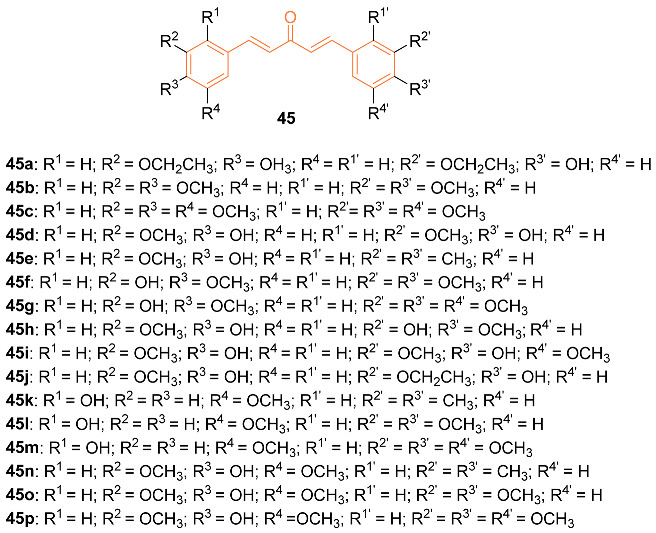



1,2,3-triazole conjugates have previously shown increased water solubility, and metabolic stability, and can participate in many interactions. With this knowledge, Mandalapu et al. utilized molecular hybridization to investigate monocarbonyl curcumin analog–1,2,3-triazole conjugates to enhance curcumin’s efficacy. Twenty-four monocarbonyl curcumin analog–1,2,3-triazole conjugates were synthesized, and though many displayed cytotoxic effects, **46** showed activity against all studied breast cancer cell lines. This was supported by IC_50_ values of 6.0 ± 1.0 µM, 10.0 ± 1.6 µM, and 6.4 ± 1.9 µM for MCF-7, MDA-MB-231, and 4TI, respectively. This compound has a good safety index and stability with the ability to cause mediated apoptosis through cellular signaling proteins in breast cancer cells. The authors believe **46** having increased activity, stability, and preferred safety can be further studied as a lead compound for a breast cancer chemotherapeutic agent [50].



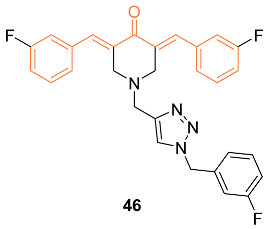



The curcumin analog dienone has previously exhibited interactions that can lead to tumor-selective toxicity. This can cause a second toxic interaction due to tumor cells having increased sensitivity to this compound. Robles-Escajeda et al., with this preliminary information, altered dienone compounds to allow binding to other cellular constituents. Three of the altered dienones, **47a**–**c**, were tested against various cancer cell lines (MDA-MB-231, MDA-MB-231-LM, HCC1419, MCF-7, MCF-10A) to determine cytotoxic effects compared to 5-fluorouracil. The dienone **47c** exhibited greater cytotoxicity for the breast cancer-derived cells. When exposed to 10-25 μM of compound **47c**, IC_50_ values of 0.88 μM, 1.11 μM, 1.46 μM, 1.83 μM, and 79.5 μM for MDA-MB-231, MDA-MB-231-LM, HCC1419, MCF-7, and MCF-10A, respectively, were found. The authors support further research through in vivo studies due to the many beneficial factors and effectiveness of dienone **47c** [51].



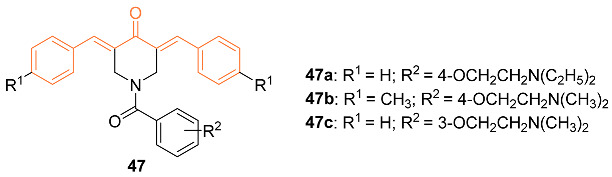



Ali et al. reported the synthesis and anti-cancer activity of the novel curcumin mimic (Z)-3-hydroxy-1-(2-hydroxyphenyl)-3-phenylprop-2-en-1-one (**48**). The mimic was tested against breast cancer cell lines MCF-7 and MDA-MB-231 as well as normal MCF-10A cell lines. The mimic had a higher IC_50_ value than curcumin at 24 h, with a value of 96.83 ± 4.87 µM for MCF-7 and 104.17 ± 5.23 µM for MDA-MB-231 compared to curcumin’s 40.72 ± 3.24 µM and 35.29 ± 4.16 µM, respectively. However, the mimic exhibited a lower IC_50_ than curcumin at both 48 and 72 h for the MCF-7 cell line, with 33.33 ± 3.50 µM for 48 h and 25.00 ± 3.71 µM for 72 h compared to curcumin’s 36.58 ± 2.31 µM and 30.15 ± 2.36 µM. In addition, the authors noted that the mimic displayed a higher selectivity for MCF-7 cells compared to normal cells than curcumin by induction of apoptosis and G2/M cell cycle arrest [52].



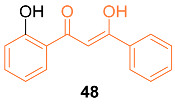



Saini et al. reported the anti-cancer activity of the novel dienone curcumin mimic **49**. An MTT assay was performed on MCF-7 and LA-7 breast cancer cells. The IC_50_ was found to be 5 µM at 24 h and 2.1 µM at 48 h for MCF-7, and 7.5 µM at 24 h and 3.1 µM at 48 h for LA-7. Cytotoxicity was not observed in the human kidney HEK-293 cell line nor the human mammary MCF-10 cell line, suggesting the selectivity of the mimic for breast cancer cell lines. The researchers concluded after studies that the inhibitor selectively binds to the minor groove of DNA, leading to DNA damage in breast cancer cells. In vivo data were obtained by implanting LA-7 cells into the mammary fat pad of rats and orally administering a 5 mg/kg body weight dose daily. Measurements were made over 21 days and showed a significant reduction in tumor size. There were no significant changes in the body weight of the rats compared to the control group as well, indicating the mimic did not have serious adverse side effects [53].



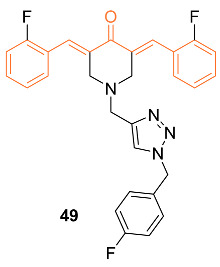



Badr et al. synthesized three curcumin analogs **50a**–**c** and tested their anti-tumor properties against the breast cancer cell lines MDA-MB-231 and MCF-7. Compound **50a** was determined to be the most effective of the three. This was supported by the IC_50_ value against MDA-MB-231 of 5 ng/mL and 10 ng/mL for MCF-7. It was also determined that **50a** did not affect the viability of normal MCF-10 breast cells. Flow cytometry was used to determine that **50a** induces growth arrest of the cancer cells by regulating the mitochondrial membrane potential [54].



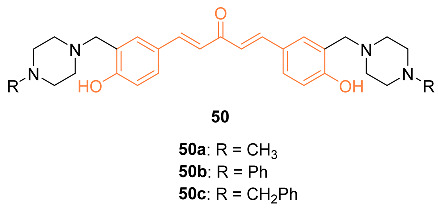



Zamrus et al. synthesized 24 curcumin analogs containing an acetone, cyclohexanone, and cyclopentanone series. Compound **52d** showed the best anti-tumor effects against the breast cancer cell lines MCF-7 and MDA-MB-231. Compound **52d** had an IC_50_ of 3.02 μg/mL for MCF-7 and 1.52 μg/mL for MDA-MB-231 breast cancer cell lines. Overall, compounds with acetone series were shown to be more selective than the other analogs. Based on structure–activity relationships, the mono-carbonyl with 2,5-dimethoxy substituted derivatives could be useful for further research on anti-cancer drugs [55].



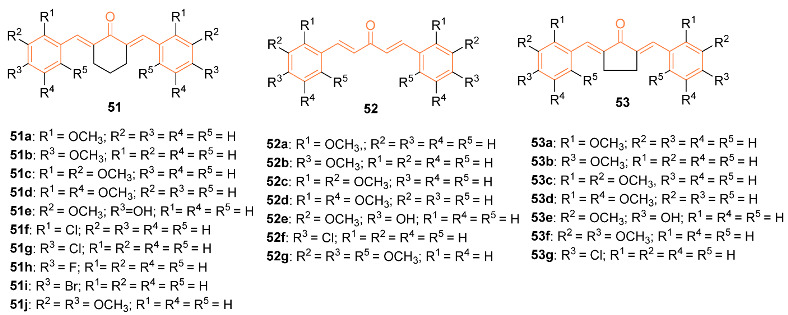



Freitas Silva et al. synthesized several curcumin hydrazide derivatives **54a**–**f** and tested their anti-tumor properties against the MCF-7 breast cancer cell line. Compound **54c** showed the most promising anti-tumor effect with an IC_50_ value of 40.49 μM compared to that of curcumin of 68.25 μM. It was found that **54c** regulates kinase proteins in breast cancer cells affecting the mitosis progression. Based on this evidence, further in vivo research should be carried out on compound **54c** as it is a promising anti-tumor agent [56].



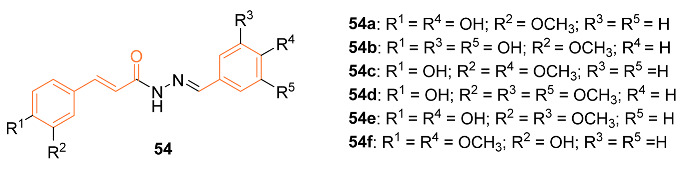



Dhongade et al. synthesized pyrazole-based curcumin derivatives **55a**–**k** and tested their anti-cancer activity against the MCF-7 breast cancer cell line by their ability to inhibit the growth of cancer cells. The IC_50_ value was used to show this inhibition with compound **55i** displaying the best anti-cancer activity compared to the other analogs with an IC_50_ of 28.75 μM [57].



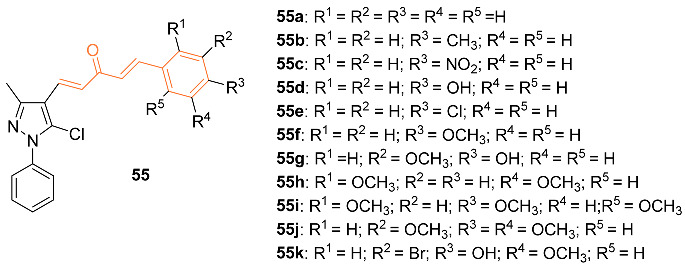



Lin et al. synthesized the line of curcumin mimics **56a**–**j** that were evaluated as potential inhibitors of PARP and HSP90 in breast cancer cells. Compounds **56a**–**c** showed noticeable proliferation in breast cancer cell lines. Furthermore, **56a** and **56c** were 200-fold more potent against MCF-7 human breast cancer cell lines than the medication olaparib. The inhibitory effect of compound **56b** on HC1937 was approximately 40 times higher than olaparib, with an IC_50_ value of 5.3 ± 3.21 μM. Additionally, **56c** showed potential as a novel anti-tumor agent as it was successful in suppressing PARP and downregulating HSP90 [58].



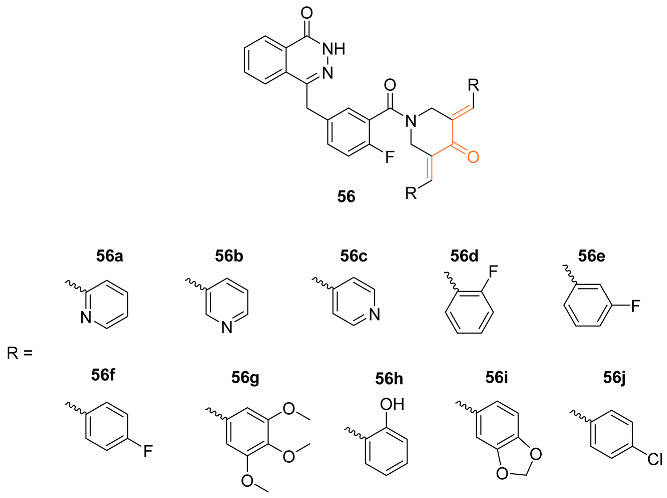



Novitasari et al. synthesized the novel curcumin analog **57** and tested the cytotoxic activity against triple-negative breast cancer cells and HER2 breast cancer cells. The authors used MTT assays with doxorubicin to determine the cytotoxic effects of this analog. They compared the effects of this analog to **58** to determine its efficacy. Compound **57** had an IC_50_ value of 6 μM after 24 h against MCF-7/HER2 cells while **58** has an IC_50_ of 16 μM [59]. Compound **58** was further investigated against the breast cancer cell line 4T1 (IC_50_ of 4 μM) and compared with curcumin (IC_50_ of 50 μM). It was found that the cytotoxic activity of compound **58** is directly related to the induction of cell cycle arrest which in turn caused an increase in intracellular ROS levels. Compound **58** showed a decrease in MMP-9 protein expression. Therefore, it can inhibit metastatic cells [59,60]. Murwanti et al. further analyzed the two curcumin analogs **43**, and **58** against breast cancer. Compound **43** had an IC_50_ value in vitro of 13.76 μg/mL against 4T1 breast cancer cells compared to 34.34 μg/mL of curcumin, while compound **58** has a value of 38.21 μg/mL. Both analogs were successful in inducing significant apoptosis at a 15 μg/mL dosage. Compound **58** has shown improvement in inducing the early apoptosis phase compared to curcumin, while compound **43** induces the late apoptosis phase better than curcumin. The authors also found that the analogs induced G2/M cell cycle arrest compared to the S phase cell cycle arrest of curcumin [61]. Further, the cytotoxic and cell migration effects of curcumin and its two analogs **43**, **35** were investigated for the two pathways associated with breast cancer: HER2 and nuclear factor kappa B. They used MTT assays, Western blotting, and immunofluorescence to generate the in vitro data. Compound **58** had an IC_50_ of 14 μM compared to curcumin (IC_50_ of 52μM), whereas compound **43** had an IC_50_ of 32 μM against HER2 breast cancer cells. Animal studies on a xenograft mouse model with triple-negative breast cancer cells also suggest compound **58** is superior for suppressing the tumor formation of metastatic breast cancer in comparison to both compound **43** and curcumin [62].



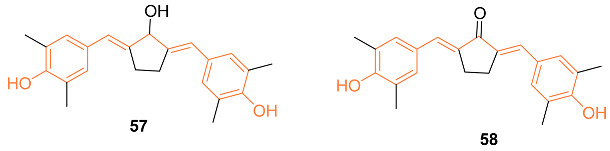



Kostrzewa et al. studied new curcumin derivatives that are PTP1B inhibitors and ROS inducers. Curcumin derivatives were analyzed for cytotoxicity against MCF-7 and MDA-MB-231 cancer cells. Compounds **59** and **60** showed the best inhibitory effect against PTP1B phosphatase which shows the potential binding of new inhibitors to the allosteric site of the enzyme. It was also discovered that by blocking the -OH group in the phenolic compounds, there was an increase in the cytotoxicity effect which caused higher levels of ROS [63].



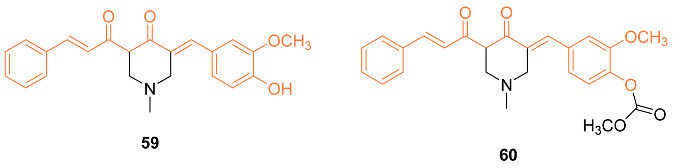



According to Adams et al., studies on **61**, a typical synthetic curcumin analog, focused on the most widely recognized signs of cell death in human breast cancer, prostate cancer, and colon cancer. The signs studied were inhibition of cancer cell proliferation, growth arrest, depolarization of mitochondrial membrane potential, activation of caspase-3, induction of PS externalization, **61**-induced redox changes, and direct interaction of **61** with GSH/Trx-1. Different concentrations of **61** (μM) were applied to MDA-MB-231 human breast cancer cells for intervals of 6–72 h. These results showed that **61** effectively inhibits active DNA synthesis by causing a G2/M cell cycle arrest, which is followed by the induction of cell apoptosis. These results stimulated additional research on **61**, which resulted in the finding that **61** causes a depolarization of the mitochondrial membrane potential by activating caspase-3 and externalizing PS. Additionally, it was discovered that **61** decreases Trx-1 and GSH levels inside cells while increasing ROS. This discovery, along with a high level of mitochondrial Ca^2+^ overload and low ATP synthesis, was what set off a series of events that led to the depolarization of the mitochondrial membrane. The findings led to the conclusion that redox-mediated induction of apoptosis plays a role in the anti-cancer activity of the novel curcumin analog **61** [64]. Further, the mechanism of action studies of **61** and curcumin in MDA-MB-231 breast cells resulted in a decrease in HIF-α1 protein levels leading to a decrease in HIF transcriptional activity. HIF-α1 levels were dose-dependently downregulated in the MDA-MB-231 cell lines after treatment with either **61** or curcumin. It was discovered that **61**′s activity was mediated by inhibiting HIF-α1 posttranscriptionally, whereas curcumin inhibited HIF-α1 gene transcription. It was also confirmed that only curcumin, not **61**, inhibited HIF-α1 transcription, suggesting that the two substances are structurally similar but functionally different. The capacity of **61**, but not curcumin, to cause microtubule stabilization in cells was another cellular effect that further distinguished the two substances. Using purified bovine brain tubulin in an in vitro assay, **61** had no stabilizing effect on tubulin polymerization, indicating that upstream signaling events may have caused the cytoskeletal disruption in cells that is brought on by **61** rather than **61** directly binding to tubulin. These findings conclude that **61** can contribute to the potent anti-cancer activity when it comes to MDA-MB-231 breast cancer cells [65].



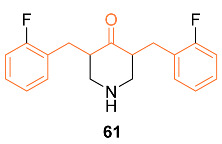



Yadav et al. investigated the cytotoxicity of 18 heterocyclic cyclohexanone curcumin analogs against MBA-MB-231, MDA-MB-468, and SkBr3 cell lines. Cells were treated for 5 days with a variety of doses of curcumin analogs for dose–response experiments. According to a study on the relationship between structure and activity, the most active pyridine heteroaromatics to date were electron-poor. In general, pyrrole-based structures had little effect on this cell line. Three of these analogs, 3,5-bis(pyridine-4-yl)-1-methylpiperidin-4-one **63a**, 3,5- bis(3,4,5-trimethoxybenzylidene)-1-methylpiperidin-4-one **63j**, and 8-methyl-2,4-bis((pyridine-4- yl)methylene)-8-aza-bicyclo [3.2.1]octan-3-one **64a**, showed potent cytotoxicity towards MBA-MB-231, MDA-MB-468, and SkBr3 cell lines with EC_50_ values below 1 μM and inhibition of NF-κB activation below 7.5 μM. The primary pharmacological candidate **63j** was similarly able to induce apoptosis in 43% of MDA-MB-231 cells after 18 h. This level of efficacy suggests that this cellular model holds promise as a prototype for the treatment of ER-negative breast cancer [66].



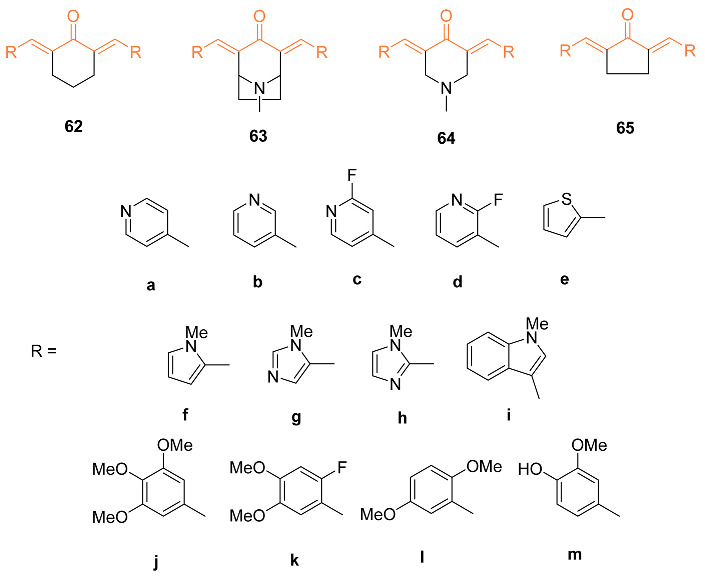



Experimental data suggest compound **66** can cause apoptosis and suppress Akt in ER-negative breast cancer cells and, due to these characteristics, this analog was subjected to additional research to determine the mechanism behind the anti-cancer activity that it exhibited in three ER-negative breast cancer cell lines: MDA-MB-231, MDA-MB-468, and SKBr3. Based on the findings, it was determined that **66** (1 μM) exhibited cell line-dependent cytotoxicity, which was brought on by an arrest in the G2/M phase of the cell cycle. In addition, **66** led to 35% of SKBr3 cells going through an apoptotic process after 48 h, which was related to a rise in cleaved caspase-3 as demonstrated by Western blotting. In addition, **66** displayed anti-angiogenic activity in vitro by inhibiting HUVEC migration and the ability of these cells to form tube-like networks. Likewise, **66** (8.5 mg/kg) was orally accessible, as its peak plasma concentration reached 0.405 g/mL 5 min after oral treatment. In SKBr3 cells, **66** inhibited HER2/neu phosphorylation and raised p27. Stress kinases JNK1/2 and p38 MAPK were transiently elevated whereas Akt phosphorylation was markedly suppressed in MDA-MB-231 and MDA-MB-468 cells after treatment with **66**. Therefore, the data give evidence that **66** has powerful anti-cancer activity and has the potential to be further developed as a medication for the treatment of ER-negative breast cancer [67].



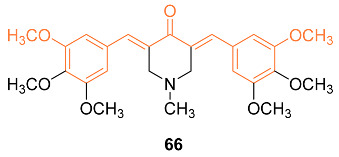



The same research group generated another analog of curcumin, **67**, which has cytotoxicity towards ER-negative breast cancer cells, and additional research into its mechanism has been conducted. The **67**-mediated cytotoxicity was investigated in MDA-MB-231, MDA-MB-468, and SKBr3 cells. These findings showed that the cell cycle arrest caused by **67** (2 μM) occurred in the G2/M phase. Additionally, **67** induced apoptosis in 40% of SKBr3 cells after 48 h due to its increased cytotoxicity. In SKBr3 cells, **67** (2 μM) decreased HER2/neu phosphorylation and increased p27, whereas, in MDA-MB-231 and MDA-MB-468 cells, it decreased Akt phosphorylation and transiently increased the stress kinases JNK1/2 and MAPK p38. Additionally, **67** demonstrated anti-angiogenic potential in vitro by preventing 46% of HUVEC migration and reducing their capacity to form tube-like networks. As a result of these findings, **67** has potent pro-apoptotic and anti-angiogenic properties in vivo and in vitro and has the potential to be developed further as a drug for the treatment of ER-negative breast cancer [68]. Another analog of **67**, **68**, was synthesized by Yamaguchi et al. and they investigated its ability to restore bone marrow cells with anti-cancer activity. The mimic was first cultured in vitro with bone marrow cells obtained from normal wild-type mice, and the results indicated that the mimic suppressed adipogenesis. In addition, the mimic stimulated osteoblastogenesis and mineralization, while osteoclastogenesis was greatly enhanced. Next, the effect of breast cancer MDA-MB-231 cells on the bone marrow was tested. When cocultured with the marrow cells, bone marrow mineralization was suppressed. However, when cocultured in the presence of the mimic, the suppression was inhibited. The cell proliferation of the mimic was also tested by culturing MDA-MB-231 metastatic cells with the mimic, and results indicated that the cells were significantly decreased with concentrations of 200 and 500 nM of the mimic [69].



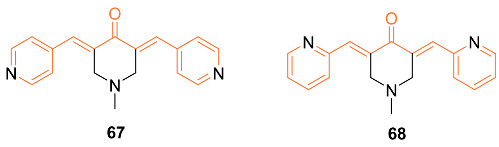



Nirgude et al. synthesized the novel curcumin derivative **69** and reported anti-cancer properties of the mimic. In vitro testing was carried out on three breast cancer cell lines MDA-MB-231, MCF-7, and T47D. The tests against MDA-MB-231 showed maximum toxicity with an IC_50_ of 0.031 μM, while MCF-7 and T47D had an IC_50_ of 0.093 and 0.138 μM, respectively. Cytotoxicity was exhibited at a nanomolar concentration, 100-fold less than curcumin, while minimal toxicity was observed against normal cells. Results also suggested that **69** promoted apoptosis in breast cancer cells, with no significant necrosis seen in any of the cells posttreatment. In vivo data were obtained by use of an EAC-induced allograft mouse model. Both **69** pre + post and posttreatment mice exhibited a significant reduction in tumor volume compared to the control group. In addition, there was no notable weight loss due to treatment in any of the mice [70].



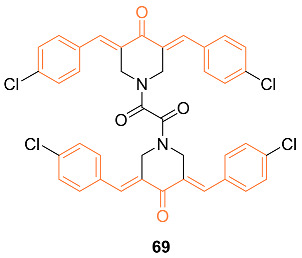



Ohori et al. reported the synthesis and evaluation of anti-cancer efficacy of six curcumin analogs **70**–**75**. In terms of their molecular structure, the novel analogs are symmetrical 1,5-diarylpentadienones with alkoxy substitutions at positions 3 and 5 on the aromatic rings. Analysis of the analogs’ effects on the expression of cancer-related genes normally impacted by curcumin revealed that some analogs promoted downregulation of B-catenin, Ki-ras, cyclin D1, c-myc, and ErbB-2 at concentrations as low as one-eighth that of curcumin. Of all the compounds tested, only analog **72** was found to inhibit DLD-1 development at a higher rate than curcumin. The IC_50_ value of analog **72** was 2.0 μmol/L, which is four times less than the IC_50_ value of curcumin (8.0 μmol/L). The ability of novel curcumin analogs to induce caspase-3-like activity was also investigated. Caspase-3 is a key component of the apoptosis pathway, which includes curcumin-induced apoptosis. In terms of caspase-3-like actions, the novel curcumin analogs were discovered to be slightly superior to curcumin. Fluorescence-activated cell sorting studies, on the other hand, clearly demonstrated that treatment with the caspase-3/caspase-8 inhibitor Z-DEVD-fmk lowered the sub-G_1_ fraction to the baseline level in either analog **70** or analog **75**. The analogs had neither harmful nor growth-stifling effects on normal hepatocytes where oncogene products are not activated. They also did not show any signs of toxicity in vivo, suggesting that they could be used as effective alternative medicines for the prevention and treatment of certain types of human cancer [71]. Pandya et al. further investigated compound **71** for its ability to interact with the G-quadruplex DNA, causing stabilization of this structure and suppressing cancer growth. Compound **71** was shown to have the highest binding affinity for the c-myc G-quadruplex DNA. Then cytotoxicity was also tested using the multicellular tumor spheroid model [72].



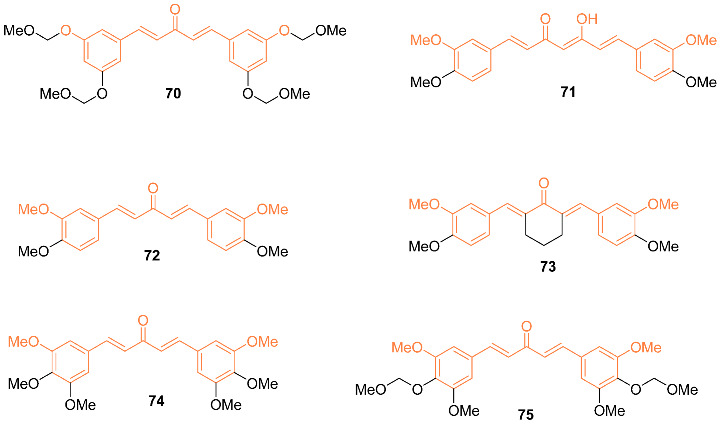



Al-Hujaily et al. studied the anti-cancer properties of curcumin analogs **76** and **77**. The study found that **77** was more efficient than curcumin and **76** in inducing apoptosis. This was assessed via annexin V/propidium iodide (PI) assay on various breast cancer and normal cells. The efficiency of **77** was ten times higher against ER-negative cells compared to ER-positive cells; however, ectopic expression of ERa rendered the ER-negative cells more resistant to **77**. Immunoblotting analysis of **77** indicated several oncoproteins known to be expressed heavily in breast cancer were suppressed/reduced in specific proteins, with the effects being pronounced with NF-κB, B-catenin, cyclin D1, Bcl-2, and survivin. The study utilized flow cytometry to investigate the analog’s effects on the cell cycle; **77** was discovered to delay the G2/M phase with a more substantial effect on ER-negative cells. The inhibitory effects of **77** were observed in breast cancer tumor xenografts developed in mice; observations include significantly reduced tumor size and triggering of apoptosis. In addition, **77** exhibited a strong capacity as an immuno-inducer by reducing the secretion of two major Th2 cytokines, IL-4 and IL-10. The analog also inhibited survivin, NF-κB, and its downstream effectors cyclin D1 and Bcl-2 while strongly upregulating p21WAF1 in both in vitro and in vivo testing. ^18^F-radiolabeled **77** was used to determine bioavailability and biodistribution in vivo; compared to curcumin, **77** showed better stability in blood and increased bioavailability [73].



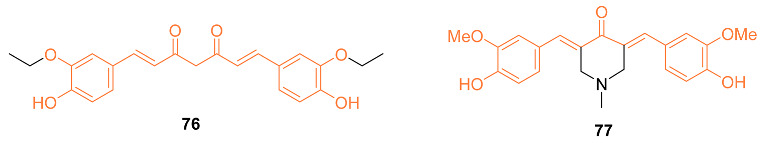



According to Faião-Flores et al., sodium 4-[5-(4-hydroxy-3-methoxyphenyl)-3-oxo-penta-1,4-dienyl]-2-methoxy-phenolate **78** was examined for anti-tumor effects in adjuvant chemotherapy for breast cancer treatment. In this study, female BALB/c mice weighing 18–23 g were used, and an MTT colorimetric assay was performed. These findings showed that curcumin and paclitaxel had higher IC_50_ values for EAT cells, 510 mM and 140 mM, respectively, whereas **78** included colorimetric cytotoxicity in EAT cells with an IC_50_ value of 99 μg/mL (284 μM). Additionally, BALB/c mice treated with a combination of paclitaxel and **78** had longer lifespans and a higher survival rate than the control groups, according to the Kaplan–Meier calculation of survival rate. These findings imply that **78** increases the survival rate, most likely by paclitaxel’s cytotoxicity being reduced, its anti-tumor potency being increased, and the tumor growth being slowed. The incidence of apoptosis in tumor cells was also increased by a combination of paclitaxel and **78** treatment and **78** treatment alone by up to 40%, which is 35% more than in the group treated with paclitaxel alone. The drug **78** is now being considered as a treatment option for breast cancer. It may also be used as an adjuvant in chemotherapy to increase the anti-tumor effects of other drugs while minimizing side effects [74].



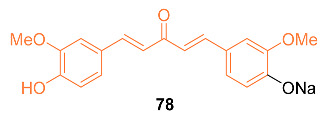



Razak et al. reported on the anti-tumor activity of the novel curcumin mimic (2*E*,6*E*)-2,6-bis(4-hydroxy-3-methoxybenzylidene) cyclohexanone **79**. In vitro tests were performed on murine 4T1 TNBC cells. The IC_50_ values for the mimic at 24, 48, and 72 h were 54.64 µM, 21.66 µM, and 13.66 µM compared to curcumin’s 81.44 µM, 48.86 µM, and 27.15 µM, respectively. In vivo data were obtained from mice with 4T1 breast cancer cells given 50 mg/kg doses of either curcumin or the curcumin mimic. After 28 days of treatment, the tumors of the curcumin group were 1.86-fold smaller and those of the mimic-treated group were 3.10-fold smaller than those of the untreated mice. In addition, neither group had significant weight loss and both the curcumin and mimic showed the ability to lower the amount of metastasis present in the mice [75].



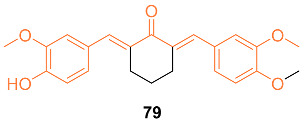



Madan et al. tested a novel curcumin analog’s (**80**) ability to kill breast cancer cells by modifying mutant p53 proteins to active wild-type p53 proteins. Compound **80** covalently binds to the mutant p53 protein which initiates a response similar to that of the wild-type p53 protein. It was also determined that this analog is exclusive to cancer cells and shows high anti-cancer efficacy in tumor models [76].



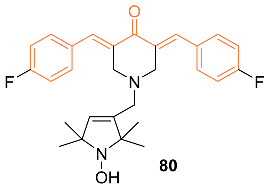



Gao et al. evaluated (*E,E*)-4,6-bis(styryl)-2-substituted amino-pyridines **81a**–**d** as P-gp modulators in the treatment of breast cancer. Studies were first carried out on P-gp-transfected LCC6 cells, which showed resistance to paclitaxel with a 90.7-fold increase compared to the parental LCC6 cells. IC_50_ values for the analogs ranged from 9.5 to 15.9 μmol/L, comparable to curcumin. The compounds showed a very strong modulating activity for P-gp in vitro, with RF values ranging from 33.3 to 86.0, compared to curcumin’s 1.3. Additionally, tests were conducted on the mRNA levels of *MDR1*, which is involved in the transport of anti-cancer agents. Additionally, **81c** displayed the most potent inhibitory activity, decreasing the mRNA level of *MDR1* to 38% and **81c** was also shown to be the most effective modulator of MDR. The Western blotting test indicated that the modulatory effect occurred at the gene level, with no change in protein level [77].



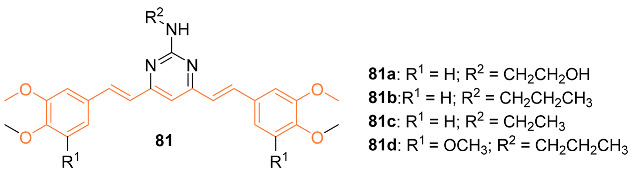



Srour et al. synthesized aspirin curcumin mimics **82a**–**p** that were tested against several cancer cell lines, including MCF-7 breast cancer cells. There was a greater activity potency by 1.19- and 1.12-fold compared to that of 5-fluoruracil by compounds **82c**,**o**. Compound **82c** had an IC_50_ of 2.653 μM and compound **82o** had an IC_50_ of 2.806 μM while 5-fluorouracil (5-FU) had IC_50_ = 3.15μM. It was also concluded that the number of methoxy groups attached to the benzylidene ring was associated with the enhancement of bioefficacies. Compound **82c** showed an arrest of the cell cycle at the G1 phase and compound **82o** showed an arrest of the cell cycle at the S phase [78].



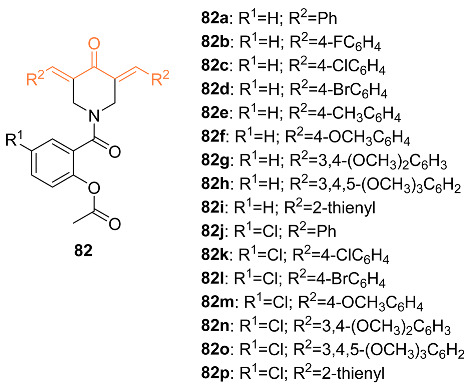



Samaan et al. synthesized over 30 heteroaromatic curcumin mimics with monoketone linkers. The most promising analogs from testing on prostate cancer (**83a**–**d**) were tested against the breast cancer cell line MDA-MB-231. Compounds **83a**–**d** exhibited IC_50_ values 7-9 times better than curcumin at 0.13 µM, 0.15 µM, 0.156 µM, and 0.097 µM, respectively, compared to curcumin’s 0.88 µM. The compounds were additionally tested against normal mammary MCF-10A cells and exhibited no apparent toxicity towards the cells up to 1 µM, with %cell survival being over 80% for all compounds [79].



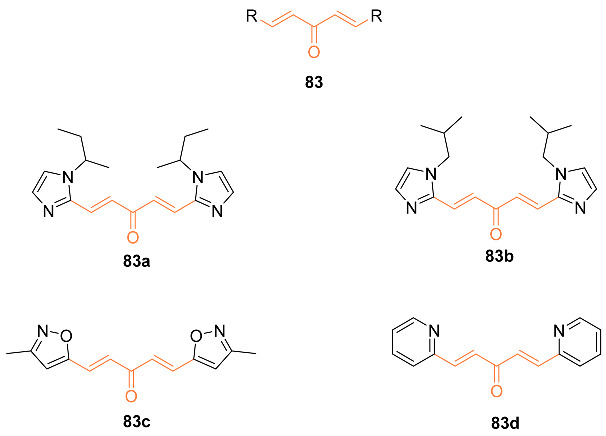



Youssef et al. synthesized several piperidone–piperazine conjugates **84a**–**k**, **85a**–**k** in the hope they act as curcumin mimics. These analogs were tested against MCF-7 breast cancer cell lines to determine their anti-proliferation abilities. Flow cytometry was performed to support the anti-proliferation abilities of these compounds. The compounds synthesized showed activity against topoisomerase IIα and I. It was shown that these compounds showed higher efficacy against topoisomerase IIα. Compound **84e** was shown to be the most potent analog out of all the synthesized compounds with an IC_50_ value of 1.148 μM compared to that of 5-fluorouracil (IC_50_ of 3.15 μM) against MCF-7 cells. It was also concluded that the 1-(2-chloroacyl)piperidin-4-one compounds with the acetyl group rather than the propyl group are more potent towards MCF-7 breast cancer cells [80].



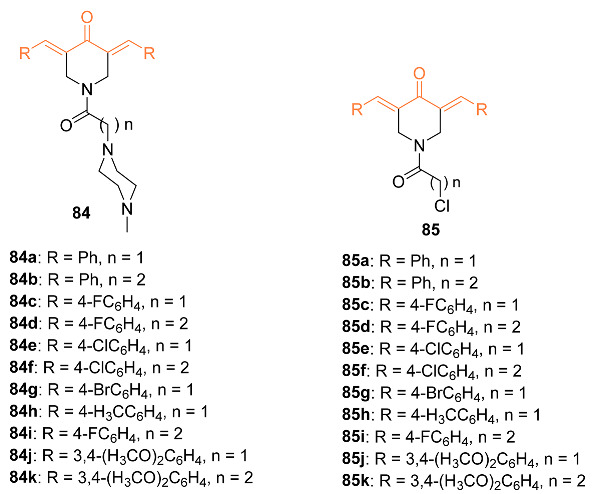



Doan et al. studied 32 asymmetric monocarbonyl analogs of curcumin that were fused with 1-aryl-1H-pyrazole. Of the 32 asymmetric monocarbonyl analogs of curcumin, approximately nine compounds exhibited growth inhibition against MDA-MB-231, with IC_50_ values ranging from 2.43 to 7.84 µM, and HepG2, with IC_50_ values ranging from 4.98 to 14.65 µM, cell lines. The selected compounds were evaluated for their selectivity toward the cancer cells and the non-cancerous LLC-PK1 cell lines using the selectivity of paclitaxel for reference; these studies concluded with six possible compounds, which were further evaluated for in vitro screening on the effects of microtubule assembly activity. Tests indicated compounds **86a**–**c** exhibited the highest microtubule destabilization effects at concentrations of 20 µm; cytotoxic studies compared to paclitaxel found that these compounds demonstrated more potent cytotoxicity towards MDA-MB-231 with three times the selectivity. IC_50_ and selectivity index values reported were 3.64 µM and 3.55 for **86b**, 5.61 µM and 2.88 for **86a**, and 5.50 µM and 2.87 for **86c**; in comparison, it was reported that curcumin had an IC_50_ of 20.65 µM and a selectivity index of 1.72 while paclitaxel had an IC_50_ of 8.79 µM and a selectivity index of 0.08. The research discovered that at concentrations of 20 µM, the compounds showed effective inhibition of microtubule assembly ranging from 40.76% to 52.03%; this suggests that the compounds may work as effective destabilizing agents. Researchers utilized MDA-MB-231 cells stained with acridine orange/ethidium bromide to observe morphological changes. At a concentration of 10.0 µM, compound **86c** exhibited a 1.33-fold increase in caspase-3 activity after 24 h, while compounds **86a**,**b** exhibited a 1.49-fold increase in caspase-3 activity after 48 h, suggesting that apoptosis was induced in the cells. Cell cycle analysis of the compounds indicated G2/M phase inhibition at concentrations of 2.5 µM for **86a**, 2.5 µM for **86b**, and 5.0 µM for **86c**. In silico modeling of the compounds revealed that the compounds’ ADMET properties were promising while also satisfying both Lipinski’s rule of five and the golden triangle rule [81].



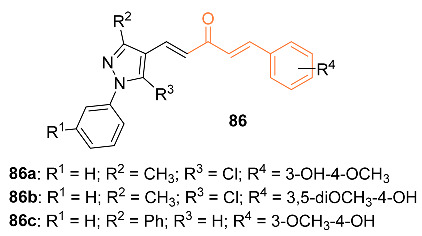



Nirgude et al. studied the curcumin derivative **87** and the drug’s effect on tumors in EAC mouse models. The study sought to explore the anti-cancer properties as a previous study of the curcumin derivative exhibited apoptotic and anti-migratory activity in MDA-MB-231 cells. It was found that **87** induced apoptosis in both in vitro and in vivo studies while also reducing tumor burden in EAC mouse models. When paired with cisplatin, an alkylating agent known to induce apoptosis in MCF-7 and MDA-MB-231 cells, the curcumin derivative exhibited a synergistic effect by sensitizing breast cancer cells to cisplatin. The study does note that it is unknown if this effect is additive or synergistic and requires further study. Changes in gene expression by **87** were evaluated through miRNA and RNA sequencing. MDA-MB-231 cells treated with **87** exhibited upregulation of TSG like GPX3, a known ROS regulator, OSGIN1, CLU, and CDH13, a known tumor suppressor gene. It was also found that the treatment of cells induced the downregulation of oncogenes. Researchers built a miRNA–mRNA interaction network on understanding gene regulation by miRNA. Data analysis using miRmapper showed miR-340, a known TSmiR in breast cancer, regulates 2.85% DE genes in MDA-MB-231 cells. This analysis also discovered that **87** restored the expression of decorin, a proteoglycan often used as a marker for good prognosis of breast cancer patients. Data analysis revealed that the curcumin derivative mediated the regulation of NF-κB and its downstream targets. The researchers observed reduced tumor burden in EAC mouse tumor models at doses of 20 mg per kg of body weight. Researchers also observed that a combination of cisplatin at 1 mg per kg of body weight and the derivative at 10 mg per kg of body weight resulted in a drastic tumor reduction in mouse models while exhibiting no apparent renal or liver toxicity [82].



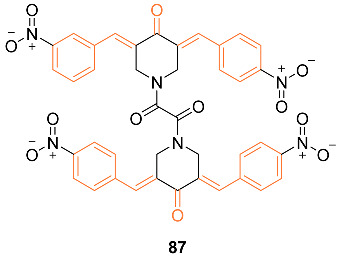



The aim of developing curcumin mimics is to overcome the potential drawbacks of curcumin while maintaining or enhancing the anti-cancer potential. While carrying out the structural modifications, it is important to study and analyze the impact of signaling pathways, and changes in antagonist or agonist effects on receptors. We have carefully collected the reported data of the mimics and compiled them together in Table 3. We observed that the mimics have effects on multiple signaling pathways but interestingly they predominately induce G2/M arrest.

### 2.4. ADME Properties

The drugability of a particular molecule can be approximately estimated using a guiding principle known as Lipinski’s rule of five (RO5). With the use of this rule, one can establish whether a biologically active chemical is likely to possess the chemical and physical qualities necessary to be orally bioavailable [83]. According to the Lipinski rule, pharmacokinetic drug features such as absorption, distribution, metabolism, and excretion are based on certain molecular qualities. These qualities include having no more than five hydrogen bond donors, a maximum of ten hydrogen bond acceptors, the mass of the molecules being less than 500 Da, and not having a value higher than five for the partition coefficient (logP). The prediction that a molecule is not capable of being taken orally is made when one or more of these conditions are violated. The name “rule of five” originates from the fact that each of the determining circumstances is a multiple of five. These physicochemical factors are linked to appropriate levels of water solubility and intestinal permeability, which are the first two crucial steps toward oral bioavailability. The RO5 does not apply to substances that are capable of being actively absorbed via the gastrointestinal tract by transport proteins. Rotatable bonds are another crucial component that should be taken into consideration. These bonds are defined as any single bond that is not in a ring and is bound to a non-terminal heavy (i.e., non-hydrogen) atom. Amide C-N bonds are not included in the count because of the large rotational energy barrier that these bonds present. The number of rotatable bonds is a significant factor in assessing the oral bioavailability of the medications since it helps measure the flexibility of the molecules. It has been discovered that the maximum number of rotatable bonds that should be present in molecules for oral medicine is ten. This is since the bioavailability of the drug would decrease as the number of rotatable bonds increases [84]. The polar surface area, also known as the topological polar surface area (TPSA), of a molecule is defined as the surface sum over all polar atoms or molecules, which predominantly consist of oxygen and nitrogen and include the hydrogen atoms that are connected to them. When calculating the PSA, the area of carbon atoms, halogen atoms, and hydrogen atoms that are linked to carbon atoms is subtracted from the total surface area of the molecule (i.e., non-polar hydrogen atoms). In other words, the PSA refers to the surface that is related to polar hydrogen atoms as well as heteroatoms, which include oxygen, nitrogen, and phosphorous atoms. It is recommended that orally active medicines that are carried by the transcellular route have a PSA that is no larger than about 140 Å. Similarly, this number ought to be tuned to PSA < 100 Å or even smaller, 60–70 Å, to ensure adequate brain penetration of CNS medicines. However, we do not concentrate on whether medicine can pass the blood–brain barrier (BBB), therefore we look at pharmaceuticals that have an ideal PSA of around 140 Å. Only central nervous system (CNS) medications can penetrate the BBB, which precludes the brain from absorbing the majority of pharmaceuticals [85]. This quality is a result of the brain capillary endothelium having tight connections that are similar to epithelial junctions. The blood–cerebrospinal fluid barrier in the choroid plexus is physically and functionally separate from the BBB. For the sake of this study, we do not want the drug to be able to penetrate the blood–brain barrier, hence we are hoping for negative values. The hERG IC_50_ value of a drug is the final key characteristic that will be covered in this discussion. The IC_50_ is the current standard for determining how effective a drug is in blocking potassium human ether-à-go-go-related gene (hERG) channels which are a well-known promiscuous drug target, and many drugs associated with torsade de pointes inhibit the *I*_Kr_ and hERG channels [86]. The ideal value for hERG pIC_50_ is less than five for consideration as a safer drug candidate. We have used the computational tool STARDROP to determine the ADME and drug-like properties, which are compiled together in Table 4 [87]. We believe the data will help the research community in the development of effective and safer drug candidates for breast cancer.

When the focus is on developing oral anti-cancer drug candidates for breast cancer, drug-like properties including Lipinski’s rule of five should be the prime criteria in addition to the rationale of drug design and synthesis. We have generated the data on common drug-like properties of the potential curcumin-based compounds compiled in this article. Most of the potential compounds have the required/desired parameter. However, some compounds (highlighted in **red**) do have anti-cancer properties but violate the recommended values of the key drug-like properties.

## 3. Clinical Trials

Curcumin has been increasingly highlighted for its ability to modulate multiple cell signaling pathways and protect against hepatic conditions, chronic arsenic exposure, and even alcohol intoxication. Due to its diverse pharmacological aspects, several clinical trials have been carried out for its use for inflammation, skin diseases, eye diseases, Alzheimer’s disease, gastrointestinal diseases, respiratory diseases, liver diseases, diabetes, cancer, depression, and anxiety [88]. A total of 12 clinical studies on curcumin have been conducted for breast cancer [89] but none of the curcumin-based compounds reached clinical trials.

## 4. Conclusions and Future Perspective

Breast cancer is the second most common cancer in women with an estimated 287,850 cases in 2022 and 51,400 deaths in the USA. Although a repertoire of many chemotherapy drugs and procedures are available for treatment, novel pharmacological agents that act through unconventional mechanisms to enhance existing therapies or kill tumor cells on their own are urgently needed. Natural products and/or natural product scaffolds are the products of interest and approach in the new drug development process. Curcumin, the active ingredient of *Curcuma longa*, has been studied extensively over the past few decades for its wide range of pharmacological properties. Curcumin has shown considerable anti-cancer effects against several different types of cancer cell lines, including prostate cancer, breast cancer, colorectal cancer, pancreatic cancer, and head and neck cancer. However, the anti-cancer application of curcumin has been limited mainly due to its low cellular uptake and poor oral bioavailability. Chemical modification of curcumin and the curcumin scaffold have been explored well to overcome the limitations and develop potential drug candidates. In this current review article, we have focused on modified curcumin and curcumin conjugates for their anti-breast cancer properties. All the compounds in this account have anti-cancer properties against breast cancer cell lines. Additionally, some compounds have been further tested in vivo using breast cancer animal models.

Despite the tremendous effort to improve the physicochemical and biological properties of curcumin, there are still several issues to be addressed regarding its bioavailability, potency, and target specificity. We believe there is still much room for improvement in curcumin and/or curcumin scaffolds in terms of selectivity for specific tumors, bioavailability, and toxicity. The computationally determined drug-like properties included in this article will enable the development of the next generation of curcumin-based drug candidates for breast cancer.

## Figures and Tables

**Figure 1 molecules-27-08891-f001:**
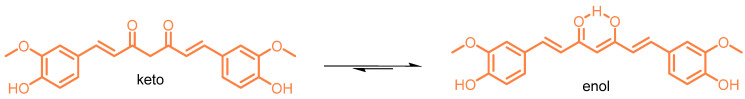
Keto-enol tautomeric forms of curcumin (CUR).

**Figure 2 molecules-27-08891-f002:**
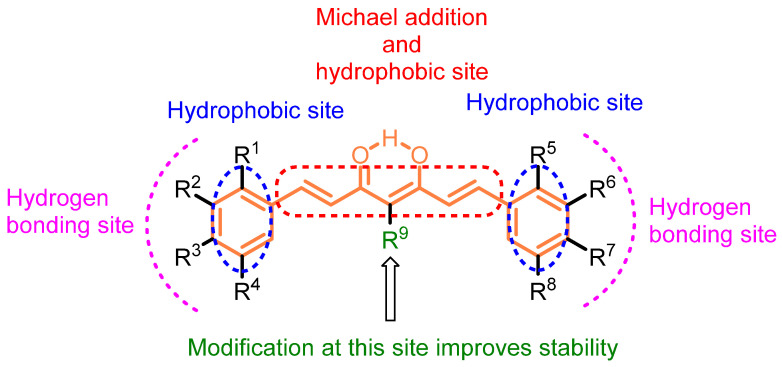
Structure of curcumin with indications of the important sites.

**Figure 3 molecules-27-08891-f003:**
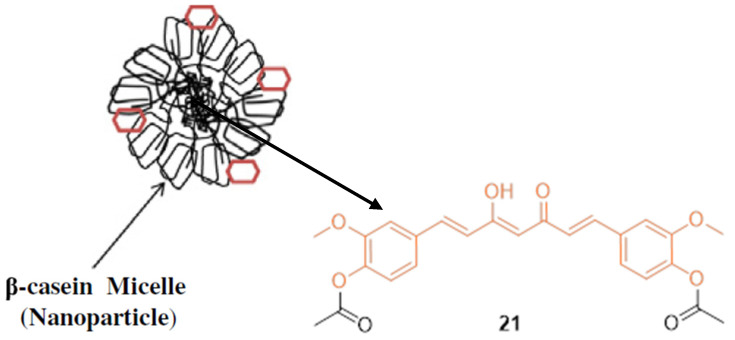
Interaction of diacetylcurcumin (DAC) with bovine β-casein.

**Table 1 molecules-27-08891-t001:** Summarized information on curcumin analogs.

Compd.	Cell Lines Tested IC_50_ (µM)	Reference Molecules IC_50_ (µM)	Pathway/ Mechanism	Regulation	Cell Cycle Arrest (Phase)	[Ref]
**1**	MCF-7 13.1 ± 1.6 MCF-7R 12.0 ± 2.0	MCF-7 (CUR) 29.3 ± 1.7 MCF-7R (CUR) 26.2 ± 1.6	NF-κB STAT3	Bcl-2↓ Bcl-X_L_↓ c-IAP-1↓ XIAP↓ NAIP↓ Survivin↓ COX-2↓	S	[18]
**2**	MCF-7 2.56 MDA-MB-231 3.37	MCF-7 (CUR) 21.22 MDA-MB-231 (CUR) 26.9	STAT3	MMP-9↓ MMP-2↓ Mcl-1↓ Cyclin D1↓ c-myc↓ Bcl-X_L_↓ Bcl-2↓ Survivin↓ VEGF↓	G1	[19]
**3**	MCF-7 30	MCF-7 (CUR) 30	NF-κB P13K/Akt	NF-κB p65↓ c-Rel↓ src↓ COX-2↓ MMP-9↓ TIMP-2↑ VEGF↓ TNFα↓ IL-1β↓ TGF-β↓ IL-8↓	NA	[21]
**4**	NA	NA	HER2 P13K/Akt ERK1/2	NA	G2/M	[22]
**5**	MCF-7 18	MCF-7 (Tamoxifen) 4	HER2	Cyclin D1↓ Survivin↓ Bcl-2↓ CDC2↓	NA	[23]
**6j**	MCF-7 15	NA	NA	Bcl-2↓	NA	[24]
**7i**	MCF-7 10	NA	NA	Bcl-2↓	NA	[24]
**8a**	MCF-7 20 *	MCF-7 (DOX) 13	NA	NA	G2/M	[25]
**9**	SKBR3 1.71 ± 0.10 MDA-MB-231 4.49 ± 0.46 MCF-7 9.09 ± 0.46 MDA-MB-453 10.42 ± 0.92	SKBR3 (Cisplatin) 3.04 ± 0.17 MDA-MB-231 (Cisplatin) 12.81 ± 0.58 MCF-7 (Cisplatin) 11.16 ± 0.60 MDA-MB-453 (Cisplatin) 15.43 ± 0.83	NA	NA	NA	[26]
**10**	MCF-7 8.84 MDA-MB-231 8.31	MCF-7 (CUR) 16.85 MDA-MB-231 (CUR) 42.01	NA	p21↑ Cyclin D1↓ CDK2↓ Bax↑ Cyto-C↑ Bcl-2↓ VEGF↓ TIMP1↑ TIMP2↑	G1	[27]
**12b**	MDA-MB-231 4.99 ± 1.02	MDA-MB-231 (Cisplatin) 6.18 ± 0.79	NA	NA	NA	[29]
**13**	MCF-7 0.317 MDA-MB-231 0.473 HS 578T 0.295 BT-549 0.693 T-47D 0.79 MDA-MB-468 0.248	NA	EGFR	NA	NA	[30]
**14**	MCF-7 0.162 MDA-MB-231 0.404 HS 578T 0.0779 BT-549 0.631 T-47D 0.793 MDA-MB-468 0.276	NA	EGFR	NA	NA	[30]
**15**	MCF-7 1.81 MDA-MB-231 1.45 HS 578T 0.524 BT-549 1.3 T-47D 2.83	NA	EGFR	NA	NA	[30]
**16**	MCF-7 2.7 ± 0.5 MDA-MB-231 1.5 ± 0.1	MCF-7 (CUR) 21.5 ± 4.7 MDA-MB-231 (CUR) 25.6 ± 4.8	NA	NA	NA	[31]
**17**	MCF-7 0.4 ± 0.1MDA-MB-231 0.6 ± 0.1	MCF-7 (CUR) 21.5 ± 4.7 MDA-MB-231 (CUR) 25.6 ± 4.8	NA	NA	NA	[31]
**18**	MCF-7 2.4 ± 1.0 MDA-MB-231 2.4 ± 0.4	MCF-7 (CUR) 21.5 ± 4.7 MDA-MB-231 (CUR) 25.6 ± 4.8	NA	NA	NA	[31]
**19b**	MCF-7 1.97 ** MDA-MB-231 3.06 ** HS 578T 5.10 ** BT-549 1.96 ** T-47D 2.81 ** MDA-MB-468 2.30 **	NA	EGFR	NA	NA	[32]

* µg/mL, ** GI_50._

**Table 2 molecules-27-08891-t002:** Summarized information on curcumin conjugates.

Compd.	Cell Lines Tested IC_50_ (µM)	Reference Molecules IC_50_ (µM)	Pathway/ Mechanism	Regulation	Cell Cycle Arrest (Phase)	[Ref]
**21**	MCF-7 22.5	MCF-7 (DAC) 26.5	NA	NA	NA	[34]
**22**	MCF-7 2.31 MDA-MB-231 5.1	MCF-7 (CUR) 40.32 MDA-MB-231(CUR) 37.87	Akt STAT3 ERK1/2	Cyclin B1↓ Cyclin A↓ Cyclin D1↓ BNIP3↑	G1/S G2/M	[35]
**25**	MCF-7 44.51± 1.74 (24 h) 42.43 ± 1.63 (48 h)	MCF-7 (Cisplatin) 56.24 ± 2.35 (24 h) 45.35 ± 1.59 (48 h)	NA	NA	NA	[36]
**26**	MCF-7 49.62 ± 2.23 (24 h) 46.32 ± 1.68 (48 h)	MCF-7 (Cisplatin) 56.24 ± 2.35 (24 h) 45.35 ± 1.59 (48 h)	NA	NA	NA	[36]
**27a**	MCF-7 6.13 ± 0.51 T47D 19.28 ± 1.71 SKBR3 42.83 ± 1.47 BT474 30.14 ± 1.42 HS-578T 55.45 ± 1.39 MDA-MB-157 40.38 ± 3.26 MDA-MB-453 42.89 ± 2.35	MCF-7(CUR) 42.89 ± 2.36 T47D(CUR) 47.91 ± 3.90 SKBR3(CUR) 6.39 ± 0.43 BT474(CUR) 6.15 ± 0.87 HS-578T(CUR) 7.96 ± 0.27 MDA-MB-157(CUR) 9.23 ± 0.11 MDA-MB-453(CUR) 6.13 ± 0.51	NA	HO-1	G2/M	[37]
**28**	MCF-7 >100 (dark) 1.9 ± 1.2 (light)	MCF-7 (Cisplatin) 28.0 ± 3.1 (Dark) ND (Light)	NA	NA	NA	[38]
**29**	MCF-7 12.0 ± 3.0	MCF-7 (CUR) 48.3 ± 2.9	NA	NA	NA	[39]
**30**	SUM 149 11.2 ± 1.30 MDA-MB-231 18.0 ± 0.41	SUM 149 (CUR) 14.0 ± 0.29 MDA-MB-231 (CUR) 25.5 ± 0.35	NF-κB	NF-κB Activity↓	NA	[40]
**31**	SUM 149 13.2 ± 1.59 MDA-MB-231 20.0 ± 0.00	SUM 149 (CUR) 14.0 ± 0.29 MDA-MB-231 (CUR) 25.5 ± 0.35	NF-κB	NF-κB Activity↓	NA	[40]
**32**	SUM 149 13.5 ± 0.88 MDA-MB-231 15.0 ± 0.85	SUM 149 (CUR) 14.0 ± 0.29 MDA-MB-231 (CUR) 25.5 ± 0.35	NF-κB	NF-κB Activity↓	NA	[40]
**38a**	MDA-MB-231 63.3 ± 0.1 (dark) 15.6 ± 0.5 (light)	NA	NA	NA	G1	[43]
**38b**	MDA-MB-231 57.3 ± 0.6 (dark) 11.3 ± 0.5 (light)	NA	NA	NA	G1	[43]
**38c**	MDA-MB-231 49.6 ± 0.8 (dark) 9.5 ± 0.3 (light)	NA	NA	NA	G1	[43]

**Table 3 molecules-27-08891-t003:** Summarized information on curcumin mimics.

Compd	Cell Lines Tested IC_50_ (µM)	Reference Molecules IC_50_ (µM)	Pathway/ Mechanism	Regulation	Cell Cycle Arrest (Phase)	[Ref]
**41**	MCF-7	NA	Cytochrome P450	Bcl-2↓ c-myc↓ Ha-ras↓ hTERT↓	NA	[46]
**42**	MCF-7 2.21 + 0.21	MCF-7 (CUR) 8.98 ± 3.21	NA	CHOP↑ Bcl-2↓ Cyclin D1↓ COX-2↓	G1/S	[47]
**43**	MCF-7 60 ± 2.04 MCF-7/Dox 21 ± 0.008	MCF-7 (CUR) 109 ± 1.915 MCF-7/Dox (CUR) 80 ± 2.39 MCF-7 (DOX) 0.4 MCF-7/Dox (DOX) 7	NA	NA	NA	[48]
**44**	MCF-7 6 ± 2.02 MCF-7/Dox 82 ± 3.09	MCF-7 (CUR) 109 ± 1.915 MCF-7/Dox (CUR) 80 ± 2.39 MCF-7 (DOX) 0.4 MCF-7/Dox (DOX) 7	NF-κB	NF-κB↓ p65↓ HER2↓	G2/M	[48]
**45e**	MCF-7 22.53	MCF-7 (CUR) 70.20	NA	NA	NA	[49]
**45p**	MCF-7 9.11	MCF-7 (CUR) 70.20	NA	NA	NA	[49]
**46**	MCF-7 6.0 ± 1.0 MDA-MB-231 10.0 ± 1.6 4T1 6.4 ± 1.9	MCF-7 (CUR) 83.1 ± 4.4 MDA-MB-231 (CUR) 75.3 ± 2.8 4T1 (CUR) 49.4 ± 1.4 MCF-7 (DOX) 0.13 ± 0.22 MDA-MB-231 (DOX) 1.6 ± 0.23 4T1 (DOX) 0.99 ± 0.13 MCF-7 (5-FU) 15 ± 0.23 MDA-MB-231 (5-FU) 19 ± 2.51 4T1 (5-FU) 23 ± 3.52 MCF-7(Nocodazole) 0.42 ± 0.32 MDA-MB-231 (Nocodazole) 1.1 ± 1.2 4T1 (Nocodazole) 1.3 ± 0.23	Akt	Akt Phosphorylation↓ PCNA↓ BAX↑ Bcl-2↓	G0/G1 S	[50]
**47a**	MDA-MB-231 0.91 MDA-MB-231-LM 0.90 HCC 1419 1.67 MCF-7 1.64 MCF-10A 41.03	MDA-MB-231 (5-FU) 7.85 MDA-MB-231-LM (5-FU) 6.03 HCC 1419 (5-FU) N/D MCF-7(5-FU) 1.7 MCF-10A (5-FU) N/D	NA	NA	NA	[51]
**47b**	MDA-MB-231 0.72 MDA-MB-231-LM 0.53 HCC 1419 1.7 MCF-7 4.4 MCF-10A 80.02	MDA-MB-231 (5-FU) 7.85 MDA-MB-231-LM (5-FU) 6.03 HCC 1419 (5-FU) N/D MCF-7(5-FU) 1.7 MCF-10A (5-FU) N/D	NA	NA	NA	[51]
**47c**	MDA-MB-231 0.88 MDA-MB-231-LM 1.11 HCC 1419 1.46 MCF-7 1.83 MCF-10A 79.65	MDA-MB-231 (5-FU) 7.85 MDA-MB-231-LM (5 FU) 6.03 HCC 1419 (5-FU) N/D MCF-7(5-FU) 1.7 MCF-10A (5-FU) N/D	Intrinsic apoptotic pathway	Caspase-3↑ Phosphatidylserine externalization↑ Mitochondrial depolarization↑	G2/M	[51]
**48**	MCF-7 25.00 ± 3.71 MDA-MB-231 37.50 ± 4.82	MCF-7 (CUR) 30.15 ± 2.36 MDA-MB-231 (CUR) 21.72 ± 3.18	ROS accumulation	ROS↑ p53↑	G2/M	[52]
**49**	MCF-7 2.1 LA-7 3.1	NA	p53	p53↑ Caspase-3↑ γH2AX foc↑	S	[53]
**50a**	MCF-7 10 ± 0 * MDA-MB-231 5 ± 0 *	NA	PI3K/Akt/mTOR NF-κB	Akt↓ mTOR↓ PKC-theta↓ Cyclin A↓ Cyclin D1↓ CDK2↓ Cyclin B1↑	G2/M	[54]
**51d**	MCF-7 8.70 ± 3.10 MDA-MB-231 2.30 ± 1.60	MCF-7 (CUR) 22.50 ± 5.50 MDA-MB-231 (CUR) 26.50 ± 1.40	NA	NA	NA	[55]
**52d**	MCF-7 3.02 ± 1.20 MDA-MB-231 1.52 ± 0.60	MCF-7 (CUR) 22.50 ± 5.50 MDA-MB-231 (CUR) 26.50 ± 1.40	NA	NA	NA	[55]
**54c**	MCF-7 40.49 ± 1.01	MCF-7 (CUR) 68.25 ± 2.59	CDKs	CCNB1↓ CDKN1A↑ p21↑	G2/M	[56]
**55i**	MCF-7 28.75 ± 0	MCF-7 (Paclitaxel) 0.35 ± 0	NA	NA	NA	[57]
**56a**	MCF-7 1.47 ± 1.49	MCF-7 (Olaparib) 232 ± 5.50	BER	PARP↓	NA	[58]
**56b**	HCC1937 5.3 ± 3.21	HCC1937 (Olaparib) 83.1 ± 8.99	BER	PARP↓	NA	[58]
**56c**	MCF-7 0.97 ± 0.13	MCF-7 (Olaparib) 232 ± 5.50	BER	PARP↓ HSP90↓ BRCA↓	NA	[58]
**57**	MCF-7/HER2+ 4.0 ± 0 4T1 1.0 ± 0	MCF-7/HER2+ (58) 90 ± 0 4T1 (58) 2.0 ± 0	MMPs	MMP-9↓	NA	[59]
**58**	4T1 38.21 ± 0 **	4T1 (CUR) 34.34 ± 0 **	ROS	NA	G2/M	[61]
**59**	MCF-7 25.30 ± 2.54 MDA-MB-231 31.86 ± 1.07	MCF-7 (CUR) 37.36 ± 1.88 MDA-MB-231 (CUR) 57.07 ± 6.23	PTEN-Akt/pAkt ROS	NA	NA	[63]
**60**	MCF-7 30.51 ± 4.11 MDA-MB-231 11.80 ± 1.43	MCF-7 (CUR) 37.36 ± 1.88 MDA-MB-231 (CUR) 57.07 ± 6.23	PTEN-Akt/pAkt ROS	PTP1B↓	NA	[63]
**61**	MDA-MB-231	NA	HIF-1	HIF-1α↓	NA	[65]
**63a**	MDA-MB-231 0.8 ± 0 *** MDA-MB-468 0.5 ± 0 SKBr3 0.6 ± 0 ***	MDA-MB-231 (CUR) 7.6 ± 0 *** MDA-MB-468 (CUR) 9.7 ± 0 *** SKBr3 (CUR) 2.4 ± 0 ***	NF-κB	NF-κB↓	NA	[66]
**63j**	MDA-MB-231 0.3 ± 0 *** MDA-MB-468 0.3 ± 0 SKBr3 0.4 ± 0 ***	MDA-MB-231 (CUR) 7.6 ± 0 *** MDA-MB-468 (CUR) 9.7 ± 0 *** SKBr3(CUR) 2.4 ± 0 ***	NF-κB	NF-κB↓	NA	[66]
**64a**	MDA-MB-231 1.1 ± 0 *** MDA-MB-468 0.6 ± 0 *** SKBr3 0.7 ± 0 ***	MDA-MB-231 (CUR) 7.6 ± 0 *** MDA-MB-468 (CUR) 9.7 ± 0 *** SKBr3 (CUR) 2.4 ± 0 ***	NF-κB	NF-κB↓	NA	[66]
**66**	MDA-MB-231 0.3 ± 0 MDA-MB-468 0.3 ± 0	NA	NF-κB MAPK p38/JNK	Akt↓ HER2/neu↓ p27↑	G2/M	[67]
**67**	MDA-MB-231 0.8 ± 0 MDA-MB-468 0.5 ± 0 SKBr3 0.6 ± 0	NA	NF-κB Akt MAPK	HER2/neu↓ p27↑ Akt↓ mTOR↓ NF-κB↓	G2/M S	[68]
**68**	MDA-MB-231	NA	MAPK/ERKNF-κB	TNF-α↑	NA	[69]
**69**	MDA-MB-231 0.031 ± 0 MCF-7 0.093 ± 0 T47D 0.138 ± 0	MDA-MB-231 (CUR) 10.53 ± 0 MCF-7 (CUR) 13.95 ± 0 T47D (CUR) 10.17 ± 0	Intrinsic apoptotic pathway MMPs	MMP1,2↓ Apaf↑ Cytochrome C↑ BAX↑ BAD↑ Bcl-2↓	G2/M	[70]
**72**	HCT116 MCF-7	NA	Caspase-3-dependent pathway	NA	NA	[71]
**77**	MCF-7 MDA-MB231 MCF-10A T-47D	NA	NF-κB	Bcl-2↓ Cyclin D1↓ P21^WAF1^↑	G2/M	[72]
**79**	4T1 13.66 ± 3.24	4T1 (CUR) 27.15 ± 2.36	p53	NA	NA	[75]
**81c**	MDA435/LCC6 70.5 ± 8.4 MDA435/LCC6MDR 59 ± 3.4	MDA435/LCC6 (CUR) 19.8 ± 3.4 MDA435/LCC6MDR (CUR) 18.0 ± 2.2	NA	NA	NA	[77]
**82c**	MCF-7 2.806 ± 0.26	MCF-7 (CUR) 16.00 ± 2.04 MCF-7 (Sunitinib) 3.97 ± 0.32 MCF-7 (5-FU) 3.15 ± 0.44	NA	NA	G1	[78]
**82o**	MCF-7 2.653 ± 0.22	MCF-7 (CUR) 16.00 ± 2.04 MCF-7 (Sunitinib) 3.97 ± 0.32 MCF-7 (5-FU) 3.15 ± 0.44	NA	NA	S	[78]
**83a**	MDA-MB-231 0.13	MDA-MB-231 (CUR) 0.88	NA	NA	NA	[79]
**83b**	MDA-MB-231 0.15	MDA-MB-231 (CUR) 0.88	NA	NA	NA	[79]
**83c**	MDA-MB-231 0.156	MDA-MB-231 (CUR) 0.88	NA	NA	NA	[79]
**83d**	MDA-MB-231 0.097	MDA-MB-231 (CUR) 0.88	NA	NA	NA	[79]
**84e**	MCF-7 1.148 ± 0.02	MCF-7 (CUR) 16.00 ± 2.04 MCF-7 (5-FU) 3.15 ± 0.44	NA	NA	G1/S	[80]
**86a**	MDA-MB-231 5.61 ± 0.24	MDA-MB-231 (CUR) 20.65 ± 0.80 MDA-MB-231 (Paclitaxel) 8.79 ± 0.96	NA	NA	G2/M	[81]
**86b**	MDA-MB-231 6.60 ± 0.16	MDA-MB-231 (CUR) 20.65 ± 0.80 MDA-MB-231 (Paclitaxel) 8.79 ± 0.96	NA	NA	G2/M	[81]
**86c**	MDA-MB-231 5.50 ± 0.22	MDA-MB-231 (CUR) 20.65 ± 0.80 MDA-MB-231 (Paclitaxel) 8.79 ± 0.96	NA	NA	G2/M	[81]
**87**	MDA-MB-231 MCF-7	MCF-7 (Cisplatin) 8.73 MCF-7 (Olaprib) 11.34 MCF-7 (DOX) 8.8 MDA-MB-231 (Cisplatin) 15.91 MDA-MB-231 (Olaprib) 11 MDA-MB-231 (DOX) 0.7	NF-κB	TSG↑	G2/M	[82]

* ng/mL, ** µg/mL, *** EC_50._

**Table 4 molecules-27-08891-t004:** ADME and drug-like properties of potential curcumin conjugates and curcumin mimic conjugates (Bold red numbers: Indicates violation of recommended key drug-like properties).

Compound	BBB	hERG pIC_50_	MW	HBA	HBD	Rotatable Bonds	TPSA	LogP
**1**	−	** 5.360 **	365.4	6	2	6	84.95	3.501
**2**	−	4.893	364.4	6	3	6	87.60	2.561
**3**	−	4.732	308.3	4	2	6	74.60	1.873
**4**	−	4.865	392.4	6	1	8	65.60	3.877
**6j**	−	** 5.149 **	** 568.6 **	10	4	10	** 173.30 **	3.733
**7i**	−	** 5.303 **	** 583.6 **	9	3	** 11 **	** 146.30 **	** 5.248 **
**8a**	−	** 6.298 **	** 727.7 **	** 12 **	3	** 11 **	** 163.00 **	** 5.107 **
**9**	+	** 5.446 **	294.3	3	2	6	57.53	3.904
**10**	−	4.853	326.3	5	2	6	75.99	2.931
**11a**	−	4.546	528.6	9	4	9	126.30	3.892
**12b**	+	** 5.690 **	377.5	4	0	10	46.61	3.173
**13**	−	** 5.679 **	** 511.6 **	8	3	9	105.8	4.547
**14**	−	** 5.669 **	468.5	7	2	8	93.81	3.986
**15**	−	** 5.169 **	391.4	7	4	6	111.5	2.837
**16**	−	4.792	354.4	5	0	8	53.99	3.724
**17**	−	4.835	414.4	7	0	10	72.45	3.092
**18**	−	** 5.113 **	414.4	7	0	10	72.45	3.051
**19b**	−	** 6.025 **	** 551.5 **	8	3	10	105.80	4.566
**20**	−	** 5.877 **	** 519.0 **	6	2	7	76.74	** 5.504 **
**22**	−	** 7.561 **	** 977.1 **	6	0	** 24 **	71.06	** 9.130 **
**23**	−	** 6.396 **	** 672.7 **	6	1	** 16 **	82.06	** 5.806 **
**24**	+	** 7.823 **	** 917.1 **	4	0	** 22 **	52.60	** 9.534 **
**27a**	−	3.786	** 600.6 **	** 12 **	4	** 18 **	** 186.10 **	1.031
**30**	−	4.946	406.4	6	2	10	85.22	4.180
**31**	−	** 5.088 **	444.5	6	1	13	74.22	3.840
**32**	−	** 5.033 **	476.5	6	3	9	96.22	4.954
**33**	−	** 5.647 **	** 915.9 **	** 18 **	** 6 **	** 24 **	** 244.40 **	2.960
**34**	−	** 5.520 **	** 860.0 **	** 17 **	3	** 24 **	** 226.50 **	2.589
**35**	−	** 5.081 **	** 862.0 **	** 18 **	3	** 24 **	** 235.70 **	2.491
**36**	−	** 5.965 **	** 969.3 **	** 11 **	2	** 24 **	** 146.30 **	** 5.476 **
**37e**	−	4.652	340.3	6	4	6	115.10	1.204
**39**	−	** 5.113 **	** 596.3 **	8	1	** 16 **	108.40	4.142
**40a**	−	3.768	** 710.4 **	** 12 **	3	** 22 **	** 166.60 **	2.543
**41**	+	4.533	224.3	2	1	3	37.30	3.845
**42**	+	** 5.647 **	480.2	3	0	8	35.53	** 6.265 **
**43**	−	** 5.084 **	352.4	5	2	4	75.99	3.335
**44**	−	4.944	380.4	5	2	4	75.99	4.032
**45o**	−	4.859	370.4	6	1	8	74.22	3.093
**46**	−	** 7.147 **	500.5	5	0	6	51.02	4.921
**47c**	−	** 6.049 **	452.5	5	0	7	49.85	4.230
**48**	+	4.280	240.3	3	2	3	57.53	4.056
**49**	−	** 7.342 **	500.5	5	0	6	51.02	4.921
**50a**	−	** 7.332 **	490.6	7	2	8	70.49	2.440
**52d**	−	** 5.120 **	354.4	5	0	8	53.99	3.660
**54c**	−	4.387	326.3	6	2	7	80.15	2.385
**55i**	−	** 5.220 **	438.9	6	0	8	62.58	4.713
**56c**	−	** 5.919 **	** 557.6 **	8	1	6	108.90	3.110
**57**	+	** 5.156 **	350.5	3	3	2	60.69	4.712
**58**	+	4.901	348.4	3	2	2	57.53	** 5.092 **
**59**	−	** 5.341 **	377.4	5	1	5	66.84	1.800
**60**	−	** 5.299 **	435.5	7	0	8	82.14	1.829
**61**	+	** 5.599 **	315.4	2	1	4	29.10	3.650
**63j**	−	** 5.494 **	495.6	8	0	8	75.69	3.279
**66**	−	** 5.338 **	469.5	8	0	8	75.69	2.592
**67**	+	4.980	291.3	4	0	2	46.09	1.628
**68**	+	** 5.001 **	291.3	4	0	2	46.09	1.628
**69**	−	** 5.903 **	** 742.5 **	6	0	7	74.76	** 6.241 **
**71**	−	4.918	396.4	6	0	10	71.06	2.089
**72**	−	5.000	354.4	5	0	8	53.99	3.724
**77**	−	** 5.437 **	381.4	6	2	4	79.23	2.570
**78**	−	4.911	348.3	5	1	7	64.99	3.084
**79**	−	** 5.264 **	380.4	5	1	5	64.99	3.962
**80**	+	** 6.944 **	464.5	4	1	4	43.78	4.737
**81c**	−	** 5.785 **	447.5	7	1	10	74.73	4.591
**82c**	−	** 5.654 **	465.5	5	0	6	63.68	** 5.017 **
**83a**	−	** 5.451 **	326.4	5	0	8	52.71	3.689
**83b**	−	** 5.343 **	326.4	5	0	8	52.71	3.689
**83c**	−	4.566	244.2	5	0	4	69.13	2.819
**83d**	+	4.399	236.3	3	0	4	42.85	1.908
**84e**	+	** 6.533 **	484.4	5	0	5	43.86	3.825
**86b**	−	** 5.151 **	424.9	6	1	7	73.58	4.329
**87**	−	** 5.462 **	** 784.7 **	** 18 **	0	** 11 **	** 258.00 **	3.740

## Data Availability

Not applicable.

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
