# Peer review of "Modified Curcumins as Potential Drug Candidates for Breast Cancer: An Overview"

_molecules, 2022, doi:10.3390/molecules27248891_

Round 1

Reviewer 1 Report

This manuscript provides a very interesting overview of curcurmin derivatives and literature testing them.

This is in light of curcurmins potential and claims made on it manyfold effects on signalling pathways and  cellular responses an essential topic. Especially since it highlights the efforts of the scientific community to convert this naturally occuring substance into an effective drug.

Unfortuneately the manuscript is does not cite a very similar review article by Yinet al. (Chem. Biodiversity, 2022; doi.org/10.1002/cbdv.202200485). That article mainly focuses on biochemical features and is in this reviewers opinion complementary.

This manuscript systematically goes though 87 ‚main‘ derivatives without apparent systematisation. It would greatly enhance the manuscript, if headlines and classification of the derivatives either according to their chemical systematics or the effects found would be introduced. Along this line, a table with systematically listing the type of tests performed (biochemical, cell lines, animal models, read out, mechanism/signalling pathway analyse) with the derivatives is desirable.

In section 2.2. a way of predicting drugability is described and the results are presented in table 1. Yet the results are not explained (neither is the meanig of bold typing in table 1). Here a more detailed explanation and discussion is needed.

Author Response

Date:               December 07, 2022

Subject:           Revised Manuscript ID molecules-2065751 

Dear Reviewer,

Thank you for reviewing our manuscript. We appreciate your helpful and valuable suggestions. We have reviewed them carefully and believe we have made all appropriate changes to our manuscript.

Comments and Suggestions for Authors

This manuscript provides a very interesting overview of curcurmin derivatives and literature testing them.

This is in light of curcurmins potential and claims made on it manyfold effects on signalling pathways and cellular responses an essential topic. Especially since it highlights the efforts of the scientific community to convert this naturally occuring substance into an effective drug.

Comment 1.

Unfortuneately the manuscript is does not cite a very similar review article by Yinet al. (Chem.Biodiversity, 2022; doi.org/10.1002/cbdv.202200485). That article mainly focuses on biochemical features and is in this reviewers opinion complementary.

Response: Thank you for your suggestion. We have now included the suggested reference in the manuscript (ref. 7).

Comment 2.

This manuscript systematically goes though 87 ‚main‘ derivatives without apparent systematization. It would greatly enhance the manuscript, if headlines and classification of the derivatives either according to their chemical systematics or the effects found would be introduced. Along this line, a table with systematically listing the type of tests performed (biochemical, cell lines, animal models, read out, mechanism/signalling pathway analyse) with the derivatives is desirable.

Response: We do appreciate your comments and valuable suggestion. We have now restructured the manuscript with categories like

  • Anti-breast cancer properties of curcumin analogs
  • Anti-breast cancer properties of curcumin conjugates
  • Anti-breast cancer properties of curcumin-mimic conjugates

Comment 3.

In section 2.2. a way of predicting drugability is described and the results are presented in table1. Yet the results are not explained (neither is the meanig of bold typing in table 1). Here a moredetailed explanation and discussion is needed.

Response: We have now explained the results and mentioned the meaning of bold values.

Reviewer 2 Report

This review has highlighted the limitation of curcumin, several modified curcumin conjugates as well as curcumin mimics were developed and studied for their anticancer properties. This review focused on the application of curcumin mimics and their conjugates for breast cancer. Authors have made a detailed literature analysis and presented it in a good and organized manner. However, I have some suggestions and corrections on the article, that are appended below.

·       Lacks graphical abstract in the manuscript.

·       Abstract is a good overview of the topic

·       Introduction: The information described in this section is appropriate and exhaustive to introduce the following sections. To increase the importance of this review, there is need to characterize curcumin in terms of its structure, therapeutic properties and as a substrate for the synthesis of valuable derivatives like tetrahydrocurcumin.

·       Add a few lines of modified Curcumins as Potential Drug Candidates against chemotherapy-induced toxicity in the introduction.

·       Under a new heading to discuss the primary molecular targets of curcumin and summarize in tabular form. Discuss the regulatory effect of curcumin through inhibiting carcinogenic miRNA and up regulating tumor suppressive miRNA.

·       There is a need to add a figure to illustrate the summary of the potential modification sites on the curcumin molecule.

·       There is a requirement for a new section to discuss the pharmacological activity of curcumin derivatives compared to curcumin and summarize in tabular forms.

·       If possible, add the available clinical trials and discuss them.

Author Response

Date:               December 07, 2022

Subject:           Revised Manuscript ID molecules-2065751 

Dear Reviewer,

Thank you for reviewing our manuscript. We appreciate your helpful and valuable suggestions. We have reviewed them carefully and believe we have made all appropriate changes to our manuscript.

Comments and Suggestions for Authors

This review has highlighted the limitation of curcumin, several modified curcuminconjugates as well as curcumin mimics were developed and studied for theiranticancer properties. This review focused on the application of curcumin mimicsand their conjugates for breast cancer. Authors have made a detailed literatureanalysis and presented it in a good and organized manner. However, I have somesuggestions and corrections on the article, that are appended below.

Comment 1.

Lacks graphical abstract in the manuscript.

Response: We have now provided the graphical abstract.

Comment 2.

Abstract is a good overview of the topic

Response: Thank you for your comment

Comment 3.

Introduction: The information described in this section is appropriateand exhaustive to introduce the following sections. To increase theimportance of this review, there is need to characterize curcumin in terms ofits structure, therapeutic properties and as a substrate for the synthesis ofvaluable derivatives like tetrahydrocurcumin..

Response: This manuscript is focused on curcumin derivatives and their anti-breast cancer properties. If the reviewer and/or editor requires we can include some synthetic components. However, with respect to your suggestion, we have incorporated a structure of curcumin indicating the important sites for variation and their impacts.

Comment 4.

Add a few lines of modified Curcumins as Potential Drug Candidates against chemotherapy-induced toxicity in the introduction.

Response: We have now included a few lines on modified Curcumins as possible Potential Drug Candidates against chemotherapy-induced toxicity.

Comment 5.

Under a new heading to discuss the primary molecular targets ofcurcumin and summarize in tabular form. Discuss the regulatory effect ofcurcumin through inhibiting carcinogenic miRNA and up regulating tumorsuppressive miRNA.

Response: We have now included tables after each section summarizing the signaling pathways, upregulation, downregulation as well as phases of cell cycle arrest.

Comment 6.

There is a need to add a figure to illustrate the summary of thepotential modification sites on the curcumin molecule.

Response: Thank you for your suggestion and we have now included figure 2 illustrating the suggested components.

Comment 7.

There is a requirement for a new section to discuss the pharmacological activity of curcumin derivatives compared to curcumin and summarize in tabular forms.

Response: As suggested, we have now included tables after each section summarizing the signaling pathways, upregulation, downregulation as well as phases of cell cycle arrest.

Comment 8

If possible, add the available clinical trials and discuss them.

Response: We have briefly discussed clinical trial studies of curcumin.

Round 2

Reviewer 1 Report

The authors have adressed the comments of this reviewer very well. Especially the new summary tables are very helpfull and explanations on curcumin scaffold are improving the manuscript majorly.

There are only two basically editorial points remaining:

1. In the section 'Clinical Trials' the references are missing (empty brackets)

2. Apparently the clinical trial have only been conducted on curcurmin and not the derivatives. This should be stated clearly in a sentence - either in the clinical trial section or in the outlook. Same would be helpfull for testing in animal models (here it seems only for #58 data are available). 

Author Response

Date:               December 08, 2022

Subject:           Revised Manuscript ID molecules-2065751 

Dear Reviewer,

Thank you for again reviewing our manuscript. We appreciate your helpful and valuable suggestions. We have reviewed them carefully and believe we have made all appropriate changes to our manuscript.

Comments and Suggestions for Authors

The authors have adressed the comments of this reviewer very well. Especially the new summary tables are very helpfull and explanations on curcumin scaffold are improving the manuscript majorly.

There are only two basically editorial points remaining:

  1. In the section 'Clinical Trials' the references are missing (empty brackets)

Response: We do apologize for the error. We have now included the reference numbers.

  1. Apparently the clinical trial have only been conducted on curcurmin and not the derivatives. This should be stated clearly in a sentence - either in the clinical trial section or in the outlook. Same would be helpfull for testing in animal models (here it seems only for #58 data are available). 

Response: We have now clearly indicated the information in the clinical trial section. Also, there are several compounds (9, 12, 27, 40, 41, 42, 47, 49, 54, 58, 67, 69, 77, 79, 87) tested in animal models. We have used the term in-vivo studies. We have now included a statement in the conclusion section.

Reviewer 2 Report

·       Most of the suggestions have been incorporated by the authors in the revised manuscript. Therefore, no issue with considering it for publication.

Author Response

Date:               December 08, 2022

Subject:           Revised Manuscript ID molecules-2065751 

Dear Reviewer,

Thank you for again reviewing our manuscript and recommending it for publication.